# SHARPNESS-AWARE DATA POISONING ATTACK

**Pengfei He**[1], **Han Xu**[1], **Jie Ren**[1], **Yingqian Cui**[1], **Shenglai Zeng**[1], **Hui Liu**[1],
**Charu C. Aggarwal**[2], **Jiliang Tang**[1]
[1]Department of Computer Science and Engineering, Michigan State University
[2]IBM T. J. Watson Research Center, New York
[1]`{hepengf1,xuhan1,renjie3,cuiyingq,zengshe1,liuhui7,tangjili}@msu.edu`
[2]`charu@us.ibm.com`

## ABSTRACT

Recent research has highlighted the vulnerability of Deep Neural Networks (DNNs) against data poisoning attacks. These attacks aim to inject poisoning samples into the models' training dataset such that the trained models have inference failures. While previous studies have executed different types of attacks, one major challenge that greatly limits their effectiveness is the uncertainty of the re-training process after the injection of poisoning samples. It includes the uncertainty of training initialization, algorithm and model architecture. To address this challenge, we propose a new strategy called "*Sharpness-Aware Data Poisoning Attack (SAPA)*". In particular, it leverages the concept of DNNs' loss landscape sharpness to optimize the poisoning effect on the (approximately) worst re-trained model. Extensive experiments demonstrate that SAPA offers a general and principled strategy that significantly enhances numerous poisoning attacks against various types of re-training uncertainty.

## 1 INTRODUCTION

The rise of machine learning (ML) models that collect training data from public sources, such as large language models (Brown et al., 2020; Radford et al.) and large visual models (Ramesh et al., 2021; Radford et al., 2021), highlights the need of robustness against data poisoning attacks (Steinhardt et al., 2017; Shafahi et al., 2018; Chen et al., 2017). Data poisoning attack refers to the threats of an adversary injecting poisoning data samples into the collected training dataset, such that the trained ML models can have malicious behaviors. For example, by injecting poisoning samples, the adversary's objective is to cause a poisoned model has a poor overall accuracy (known as *un-targeted attacks* (Steinhardt et al., 2017; Li et al., 2020b; Ren et al., 2022)), or misclassifies a specified subset of test samples (known as *targeted attacks* (Shafahi et al., 2018; Zhu et al., 2019)). Additionally, in *backdoor attacks* (Chen et al., 2017; Saha et al., 2020; Tran et al., 2018), the adversary aims to create "backdoors" in the poisoned model such that the model gives a specific output as the adversary desires if the backdoor trigger is presented, regardless of the actual input.

Many poisoning attacks in Deep Neural Networks (DNNs) face a common obstacle that limits their effectiveness, which is the uncertainty of the re-training process after the injection of poisoning samples. This challenge is also highlighted in previous studies such as (Schwarzschild et al., 2021; Zhu et al., 2019; Huang et al., 2020; Geiping et al., 2020). In particular, most existing methods (as reviewed in Section 2 and Appendix B) generate poisoning samples based on the effect of only one victim model (Shafahi et al., 2018; Souri et al., 2022). However, on the poisoned dataset, the re-trained model may converge to a very different point due to the re-training uncertainty, such as training algorithm, model initialization and model architectures. As a consequence, the injected poisoning samples could lose their efficacy and the poisoning effect is compromised. To overcome this difficulty, recent methods (Geiping et al., 2020; Huang et al., 2020) have introduced the *Ensemble and Re-initialization (E&R) Strategy:* which proposes to optimize the average poisoning effect of multiple models with various model-architectures and initializations. However, this method tends to be computationally inefficient (Huang et al., 2020). In general, if we consider the poisoning attacks as general bilevel optimization problems (Bard, 2013; Colson et al., 2007) (Eq. 2 in Section 3), poisoning problems in DNNs can be categorized as "multiple inner minima" bilevel optimization problems (Sow et al., 2022; Liu et al., 2020; Li et al., 2020a). Prior works (Sow et al., 2022; Li

et al., 2020a; Liu et al., 2020; 2021) have demonstrated the theoretical difficulty and complexity of solving "multiple inner minima" problems.

In this work, we introduce a new attack strategy, *Sharpness-Aware Data Poisoning Attack (SAPA)*. In this method, we aim to inject poisoning samples, which **optimizes the poisoning effect of the approximately "worst" re-trained model**. In other words, even the worst re-trained model (which achieves the relatively worst poisoning effect) can have a strong poisoning effect. Furthermore, we find that this strategy can be successfully accomplished by tackling the victim model's loss landscape sharpness (Foret et al., 2020). Notably, the loss landscape sharpness is more frequently used to explain the generalization characteristic of DNNs. In our work, we show the possibility to leverage the algorithms of sharpness (such as (Foret et al., 2020)) to advance poisoning attacks. Through experimental studies, we show that our proposed method **SAPA is a general and principle strategy** and it can improve the performance of various poisoning attacks, including targeted attacks, un-targeted attacks and backdoor attacks (Section 5.1, 5.2& 5.3). Moreover, we show SAPA can also effectively improve the poisoning effect under various re-training uncertainties, including for re-training algorithm as well as model architectures (Section 5.4). Compared to the E&R strategy (Huang et al., 2020), we show SAPA is more computational efficient (Section 5.5).

## 2    RELATED WORK

### 2.1    END-TO-END DATA POISONING ATTACKS

Data poisoning attacks (Biggio et al., 2012; Steinhardt et al., 2017) refer to the adversarial threat during the data collection phase of training ML models. These attacks manipulate the training data so that the trained models have malicious behaviors. Common objectives of poisoning attacks include the purposes of causing a poisoned model to have a poor overall accuracy (*un-targeted attacks* (Steinhardt et al., 2017; Li et al., 2020b; Ren et al., 2022)), misclassifying a specified subset of test samples (*targeted attacks* (Shafahi et al., 2018; Zhu et al., 2019)), or insert backdoors (*backdoor attacks* (Chen et al., 2017; Saha et al., 2020; Tran et al., 2018)). Notably, in this work, we focus on the poisoning attacks in an ***"end-to-end"*** manner, which means that the victim model is trained on the poisoned dataset from scratch. It is a practical setting for poisoning attacks as the attackers cannot take control of the re-training process.

In general, the feasibility of poisoning attacks highly depends on the complexity of the victim model. For linear models, such as logistic regression and SVM, there are methods (Biggio et al., 2012; Steinhardt et al., 2017; Koh et al., 2022) that can find solutions close to optimal ones. For DNNs, the exact solution is generally considered intractable, due to the complexity and uncertainty of the re-training process. In detail, in the end-to-end scenario, the victim model can be trained with a great variety of choices, including various choices of model architectures, initialization, and training algorithms, which are hard to thoroughly consider while generating the poisoning samples. Faced with this difficulty, many works (Shaban et al., 2019) focus on the transfer learning setting (Shafahi et al., 2018; Zhu et al., 2019), where they loosen the "end-to-end" assumption. They assume the attacker has knowledge of a pre-trained model, and the victim model is fine-tuned on this pre-trained model. However, these attacks are usually ineffective in the end-to-end setting. Recent works such as Gradient Matching (Geiping et al., 2020) make a great progress under end-to-end setting. However, these attacks still face obvious degradation when there are re-training variations such as model architectures. To further improve the attacks, Huang et al. (2020) devise the "ensemble and re-initialization" strategy to take several victim models into consideration. However, it is time and memory-consuming, and cannot sufficiently cover the re-training possibilities. In Appendix B, we provide a more comprehensive review of existing poisoning attacks. In Appendix B.2, we introduce some poisoning attacks which consider the scenario different from end-to-end scenario.

### 2.2    LOSS LANDSCAPE SHARPNESS

In this paper, our proposed method involves calculating and optimizing the loss landscape sharpness of DNNs. The notion of the loss landscape sharpness and its connection to generalization has been extensively studied, both empirically (Keskar et al., 2016; Dinh et al., 2017) and theoretically (Dziugaite & Roy, 2017; Neyshabur et al., 2017). These studies have motivated the development of methods (Izmailov et al., 2018; Chaudhari et al., 2019) that aim to improve model generalization by manipulating or penalizing sharpness. Among these methods, Sharpness-Aware Minimization (SAM) (Foret et al., 2020) has shown to be highly effective and scalable for DNNs. In this paper, we explore the use of sharpness for data poisoning attacks.

## 3 PRELIMINARY

In this section, we introduce the definitions of the (loss landscape) sharpness, as well as the formulations of several most frequently studied data poisoning attacks. We start by providing some necessary notations. In this paper, we focus on classification tasks, with input $x \in \mathcal{X}$ and label $y \in \mathcal{Y}$ following the distribution $D$ which is supported on $\mathcal{X} \times \mathcal{Y}$. Under this dataset, a classification model $f(\cdot; \theta) : \mathcal{X} \to \mathcal{Y}$ is trained on the training set $D_{tr} = \{(x_i, y_i), i = 1, ..., n\}$, whose $n$ data samples follow $D$. The model parameter $\theta$ is from the parameter space $\Theta$. We define the loss function as $l(f(x; \theta), y)$, and the (training) loss as $L(\theta; D_{tr}) = \frac{1}{n} \sum_{i=1}^{n} l(f(x_i; \theta), y_i)$.

### 3.1 LOSS LANDSCAPE SHARPNESS

We follow the precedent work (Foret et al., 2020) to define the loss landscape sharpness (refer to as "sharpness" for simplicity) as in Eq. 1. It measures how quickly the model's training loss can be increased by moving its parameter to a nearby region. Following Eq. 1, it calculates the training loss increase after the model parameter $\theta$ is perturbed by $v$, whose $l_p$ norm is constrained by $||v||_p \le \rho$:

$$S^{\rho}(\theta; D_{tr}) = \max_{||v||_p \le \rho} \left[ L(\theta + v; D_{tr}) \right] - L(\theta; D_{tr}). \tag{1}$$

In this work, we focus on the definition of sharpness as in Eq.1. Note that there are other sharpness definitions (Andriushchenko & Flammarion, 2022), we will investigate them as one future work. Given this definition of sharpness, previous works (Foret et al., 2020; Wen et al., 2023) build the theoretical and empirical connections between sharpness and model generalization. Generally, a smaller sharpness indicates a better generalization performance.

### 3.2 DATA POISONING ATTACKS

In this subsection, we give the definition of poisoning attacks we studied. Given a training set $D_{tr}$ with $n$ samples, we assume that the attacker selects a subset $D_q$ from $D_{tr}$ which takes an $\epsilon \in [0, 1]$ percentage of $D_{tr}$, and replaces it with a poisoning set $D_p$. Usually, the samples in $D_p$ are from modifying samples in $D_q$. During the re-training stage, the model is trained from scratch on the perturbed training set $(D_{tr} - D_q) \cup D_p$ and we denote it as "$D_{tr} + D_p$" for simplicity. Although different attacks may have different purposes and formulations, we give a general formulation as in Eq.2: the attacker aims to find the poisoning samples $D_p$, such that the trained model has an optimized malicious behavior towards data samples from a victim set $D_T$:

$$\min_{D_p} \; Q(\theta^*, D_T), \;\; \text{s.t.} \;\; \theta^* = \arg\min_{\theta} L(\theta; D_{tr} + D_p) \tag{2}$$

where $\mathcal{C}$ denotes the constraint of the poisoning samples. Notably, the design of "***adversarial objective***" $Q(\cdot)$ and "***victim set***" $D_T$ is based on the purpose of the adversary. By giving different formulations of $Q(\cdot)$ and definitions of $D_T$, the attacker can achieve various adversarial goals. For example, in targeted attacks (Shafahi et al., 2018; Geiping et al., 2020), they aim to cause the model to misclassify a certain test sample (or set of test samples). Thus, they define the victim set $D_T = \{(x_i^{vic}, y_i^{vic})\}_{i=1}^{k}$ and the adversarial objective as:

$$Q_t(\theta^*, D_T) = \sum_{(x_i, y_i) \in D_T} l(f(x_i; \theta^*), y^{obj}), \tag{3}$$

where $y^{obj}$ is a designated class different from $y^{vic}$. In Appendix B.1, we provide a detailed introduction to the threat models of two other types of poisoning attacks, such as backdoor attacks (Wu et al., 2022) and un-targeted attacks (Huang et al., 2021).

## 4 METHOD

In this section, we introduce our proposed method Sharpness-aware Data Poisoning Attack (SAPA). In Section 4.1, we will first introduce SAPA in general form. In Section 4.2, we introduce how to achieve SAPA in targeted and backdoor attacks. In Section 4.3, we introduce SAPA in untargeted attacks. From our discussion, we show the idea of leveraging sharpness is general and lightweight to be incorporated into various types of poisoning attacks.

### 4.1 THE OBJECTIVE OF SAPA

As discussed in Section 1 & 2, one major challenge for poisoning attacks originates from the existence of multiple solutions during re-training. Therefore, we first define the ***Poisoned Model Space***,

denotes as $\Theta_p$, which is the set of all models that are trained on a poisoned dataset $D_{tr} + D_p$ and have a small training loss:

$$\Theta_p = \{\theta : L(\theta; D_{tr} + D_p) \leq \tau\} \tag{4}$$

In practice, the space $\Theta_p$ composes the models that are trained from different training algorithms, initializations and architectures. In the space $\Theta_p$, we desire to find the ***Worst-case Poisoned Model***, which is the model that has the worst poisoning effect $\theta' = \arg\max_{\theta \in \Theta_p} Q(\theta; D_T)$. Ideally, we aim to optimize the poisoning effect of $\theta'$ to overcome the re-training uncertainty:

$$\min_{D_p} Q(\theta'; D_T), \tag{5}$$

$$\text{where } \theta' = \arg\max Q(\theta; D_T), \text{ s.t. } L(\theta; D_{tr} + D_p) \leq \tau. \tag{6}$$

By solving this problem, we can find poisoning samples $D_p$ such that the worst poisoned model $\theta'$ can have an optimized poisoning effect (with a small $Q(\theta'; D_T)$). Therefore, for the models that are re-trained on the poisoned dataset, the poisoning effect is likely to persist, and our experiments in Section 5 provide empirical evidence. Admittedly, it is also hard to solve the above problem exactly. As a solution, we propose to approximate the term $Q(\theta'; D_T)$, by leveraging the existing notion of model sharpness (Foret et al., 2020). In detail, given a fixed $D_p$, we approximate $Q(\theta'; D_T)$ by:

$$Q(\theta'; D_T) \approx \max_{\|v\|_p \leq \rho} Q(\theta^* + v; D_T), \text{ where } \theta^* = \arg\min_{\theta \in \Theta} L(\theta; D_{tr} + D_p) \tag{7}$$

In details, we first trained the model $\theta^*$ on $(D_{tr} + D_p)$ (i.e., via ERM), so that $\theta^*$ satisfies the constraint $L(\theta^*; D_{tr} + D_p) \leq \tau$. Then, we locally maximize the term $Q(\theta^* + v)$ by perturbing the model parameter $\theta^*$ with a vector $v$ (which has a limited norm $\|v\|_p \leq \rho$). In this way, the perturbed model $\theta^* + v$ has a worse poisoning effect compared to $\theta^*$, but it is still likely to fall into the space $\Theta_p$ (because $\theta^* + v$ is not far away from $\theta^*$). Remarkably, the form of the objective Eq. 7 resembles the definition of model sharpness in Eq. 1, while we focus on adversarial loss which is distinct from the sharpness on the standard loss landscape. Therefore, we call the term in Eq. 7 as ***Sharpness-aware adversarial objective***, and we desire to find $D_p$ to optimize this objective:

$$\min_{D_p} \max_{\|v\|_p \leq \rho} Q(\theta^* + v; D_T), \text{ s.t. } \theta^* = \arg\min_{\theta \in \Theta} L(\theta; D_{tr} + D_p) \tag{8}$$

In general, the formulation of SAPA can be seen as a modification of existing attacks: it replaces the adversarial objective (Eq. 2) of existing attacks with Sharpness-aware adversarial objective. Intuitively, compared with the traditional adversarial objective (Eq. 2), in SAPA, the perturbation on the inner minima $\theta^*$ with $v$ enables the algorithm to escape from the poisoned models where the poisoning effect is unstable. Thus, SAPA has the potential to induce the re-trained models to have a stable poisoning effect under the uncertainty of the re-training process. Notably, in our proposed approximation in Eq.7, we locally search for a worst model with the same model architecture of a trained model $\theta^*$, which does not explicitly consider the uncertainty of different architectures. However, empirical results in Section 5.4 demonstrate that this strategy can also help improve the poisoning effect under if there is model architecture shift.

## 4.2 SAPA IN TARGETED /BACKDOOR ATTACKS

In this subsection, we take one representative algorithm (Geiping et al., 2020) in targeted attacks as an example, to show SAPA can be incorporated into existing attacks without requiring additional attacker's knowledge or computational overhead. In the work Gradient Matching (Geiping et al., 2020), they assume the attacker is targeting on a victim set $D_T$, and the attacker has other information including a pre-trained model $\theta^*$, which is trained on (or part of) the clean training set $D_{tr}$ and a loss function $L(\cdot)$. In their attack, they solve the following problem to find the set $D_p$ satisfying:

$$\arg\min_{D_p} \left(1 - \frac{\langle \nabla_\theta Q(\theta^*; D_T), \nabla_\theta L(\theta^*; D_{tr} + D_p)\rangle}{\|\nabla_\theta Q(\theta^*; D_T)\| \cdot \|\nabla_\theta L(\theta^*; D_{tr} + D_p)\|}\right). \tag{9}$$

It finds a poisoning set $D_p$, such that the gradient of the training loss, $\nabla_\theta L(\theta^*; D_{tr} + D_p)$, has a maximized alignment with $\nabla_\theta Q(\theta^*; D_T)$. In this way, during the re-training process, the model is updated by $\nabla_\theta L(\theta^*; D_{tr} + D_p)$ and likely to minimize $Q(\theta^*; D_T)$ and achieve the attack goal successfully. In our method SAPA, if we denote the sharpness-aware poisoning objective as $Q^S(\theta^*; D_T) = \max_{\|v\|_p \leq \rho} Q(\theta^* + v; D_T)$ for simplicity, the objective of SAPA is to solve:

$$\arg\min_{D_p} \left(1 - \frac{\langle \nabla_\theta Q^S(\theta^*; D_T), \nabla_\theta L(\theta^*; D_{tr} + D_p)\rangle}{\|\nabla_\theta Q^S(\theta^*; D_T)\| \cdot \|\nabla_\theta L(\theta^*; D_{tr} + D_p)\|}\right), \tag{10}$$

which directly replaces the adversarial objective in Eq.9 to be sharpness aware adversarial objective in Eq.10. In our algorithm to calculate the term $\nabla_\theta Q^S(\theta^*; D_T)$, we follow the approach in the previous work (Foret et al., 2020) to first approximate $Q^S(\theta^*; D_T)$ by leveraging a first-order method for $\hat{v}$:

$$\hat{v} = \rho \cdot \text{sign}\left(\nabla_\theta Q(\theta^*; D_T)\right) |\nabla_\theta Q(\theta^*; D_T)|^{q-1} \left(\|\nabla_\theta Q(\theta^*; D_T)\|_q^q\right)^{1/p}, \tag{11}$$

where $1/p + 1/q = 1$ and we consider $p = 2$ as illustrated in (Foret et al., 2020) if not specified. Then, we can have the approximation to calculate $\nabla_\theta Q^S(\theta^*; D_T)$ via replacing $\theta^*$ with $\theta^* + \hat{v}$:

$$\nabla_\theta Q^S(\theta^*; D_T) \approx \nabla_\theta Q(\theta; D_T)|_{\theta=\theta^*+\hat{v}} \tag{12}$$

In this way, by fixing $\nabla_\theta Q^S(\theta^*; D_T)$, we can solve Eq.10 to find the poisoning samples $D_p$ via gradient descent. Notably, the computation in Eq.11&12 does not require additional attacker's knowledge compared to original Gradient Matching. In Appendix C.1, we provide the detailed algorithm of SAPA in Gradient Matching. We also analyze its computational efficiency to show SAPA does not introduce great computational overhead. Besides, the similar strategy of SAPA and algorithm can also be applied in backdoor attacks, given the similarity between the objectives and viable solutions of targeted attacks and backdoor attacks (see more discussions in Appendix C.2).

### 4.3 SAPA IN UN-TARGETED ATTACKS

For untargeted attacks, similar strategy to replace the (investigated) model with a worse model during generating the poisoning samples also helps. Note that conducting un-targeted attacks for DNN models with a small poisoning budget $\epsilon$ is usually considered to be a hard problem (Muñoz-González et al., 2017). Existing feasible solutions (Fowl et al., 2021; Ren et al., 2022) are motivated from a "data protection perspective" (which are often referred as "un-learnable examples"). In detail, they perturb the whole (or at least a large portion of) training set, such that the trained classifiers cannot effectively learn the true data distribution and have a low accuracy on the true data. Take the method **Error-Min** (Huang et al., 2021) as an example, to induce the model to have a low accuracy, it generates "shortcut" perturbations $\delta_i$ for each training sample to solve the following bilevel problem, such that the model trained on this perturbed dataset has a minimized loss:

$$\min_{\theta \in \Theta} \min_{\{\delta_i\}_{i=1,\ldots,n}} \left[ \sum_{(x_i,y_i) \in D_{tr}} l\big(f(x_i + \delta_i; \theta), y_i\big) \right], \tag{13}$$

As a consequence, the found $\delta_i$ via solving Eq. 13 has patterns with a strong correlation to the labels $y_i$. The models trained on the perturbed dataset can predict $(x_i + \delta_i)$ to be $y_i$ mostly based on the information of $\delta_i$, and prohibit the model to learn useful knowledge from the clean samples in $D_{tr}$. In the original algorithm proposed by (Huang et al., 2021), the problem in Eq.13 is solved by updating the model parameter $\theta$ and training set perturbation $\{\delta_i\}_{i=1}^n$ alternatively. See Algorithm 1 (without **SAPA**), they first update the model parameter $\theta$ with $M$ steps (step 1) and then update data perturbation $\delta_i$ for $T$ steps (step 3). For our method, similar to the high-level idea as in targeted attacks, SAPA introduces one more step (step 2) to replace the model $\theta$ with $\theta + \hat{v}$ during generating the poisoning samples. Notably, a similar strategy can also be adapted to other un-targeted attacks, i.e., Error-Max Fowl et al. (2021). We provide its detailed algorithm in Appendix C.

## 5 EXPERIMENT

In this section, we conduct experiments to show that SAPA can improve existing methods when it is applied to targeted, backdoor and un-targeted attacks in Section 5.1, 5.2 and 5.3, respectively. For all methods in experiments, we obtain the poisoning samples via ResNet18 models. We report the results when the re-training is under ResNet18 and ResNet50, to have a thorough comparison with previous works (many baseline methods are also evaluated in the same setting). In Section.5.4 we evaluate the robustness of SAPA under various re-training settings including different training algorithms (including some defense strategies) and several more model architectures. In Section 5.5, we compare SAPA with baselines incorporating *ensemble* and *re-initialization* technique to illustrate the superiority of SAPA in both effectiveness and efficiency. Through this section, we focus on image classification tasks on benchmark datasets CIFAR10 and CIFAR100. Meanwhile, we provide additional empirical results of the dataset SVHN in Appendix D. For our method SAPA, we set the radius $\rho$ (Eq. 8 in Section 4) to be 0.05 in all experiments. We provide detailed implementation in Appendix C for all SAPA-based algorithms[1].

[1]Code is available in https://github.com/PengfeiHePower/SAPA

---

**Algorithm 1:** Error-min**+SAPA**

---

**Input** : Clean training set $\{(x_i, y_i)\}_{i=1}^n$; Optimization step $T$ and $M$; epochs $E$.
**Output:** Sample-wise perturbation $D_p = \{\delta_i\}_{i=1}^n$
Randomly initialize perturbation $D_P = \{\delta_i\}_{i=1}^n$
**for** *epoch in* $1, ..., E$ **do**
    1. **for** *m in* $1, ..., M$ **do**
        Update $\theta$ via minimizing $\sum_{(x_i, y_i) \in D_{tr}} l(f(x_i + \delta_i; \theta), y_i)$
    **end**
    2. ***SAPA:*** Fix $\theta$, $D_P$, find worst direction $\hat{v}$ to maximize $\sum_{(x_i, y_i) \in D_{tr}} l(f(x_i + \delta_i; \theta + \hat{v}), y_i)$
    3. **for** $t = 1, ..., T$ *steps* **do**
        Update $\delta_i$ via gradient descent on the fixed model $f(\cdot; \theta + \hat{v})$
    **end**
**end**

---

## 5.1 PERFORMANCE OF SAPA IN TARGETED ATTACKS

**Experiment Setup**. In this experiment, we focus on targeted attack goal to cause the poisoned model to misclassify one victim sample. During attacks, under each dataset, we assume that the attacker randomly chooses a small proportion of training dataset with "*poisoning ratio*" $\epsilon = 1\%, 0.2\%$, and inserts unnoticeable perturbations (whose $l_\infty$ norm is limited by "*perturbation budget*" $16/255, 8/255, 4/255$) on each of them. After crafting the poisoned dataset, the model is randomly initialized and re-trained from scratch, via SGD for 160 epochs with an initial learning rate of 0.1 and decay by 0.1 at epochs 80 and 120. For each setting, we repeat the experiment 50 times and report the average performance. Multiple-target attacks are also considered in Appendix D. More details of the implementation can be found in Appendix.C.1.

**Baselines**. We compare SAPA with representative baselines, including Bullseye(Aghakhani et al., 2021), Poison-Frog(Shafahi et al., 2018), Meta-Poison (Huang et al., 2020), and Grad-Match (Geiping et al., 2020). Notably, our method is incorporated to the method Grad-Match (Geiping et al., 2020). For other baselines, MetaPoison leverages meta learning (Vilalta & Drissi, 2002) to unroll the training pipeline and ensembles multiple models. Poison-frog and Bullseye, also known as "feature collision", generate poisoning samples with representations similar to those of the victim. In Appendix B, we provide detailed discussions of these baselines.

**Performance comparison**. In Table 1 we report the "Success Rate" which is the probability that the targeted sample is successfully classified to be the designated wrong label $y^{obj}$. From the results, we can see that SAPA consistently outperforms baselines. The advantage of SAPA is obvious, especially when the attacker's capacity is limited (i.e., the perturbation budget is $8/255$ or $4/255$). Additional results on ResNet50 can be found in Appendix D.

Table 1: Success Rate in Targeted Attacks(with standard error reported).

| | | 16/255 | | 8/255 | | 4/255 | |
|---|---|---|---|---|---|---|---|
| | | $\epsilon = 1\%$ | $\epsilon = 0.2\%$ | $\epsilon = 1\%$ | $\epsilon = 0.2\%$ | $\epsilon = 1\%$ | $\epsilon = 0.2\%$ |
| **CIFAR10** | **Bullseye** | 3.7±1.6 | 1.1±0.4 | 1±0.3 | 0±0 | 0±0 | 0±0 |
| | **Poison-Frog** | 1.3±0.8 | 0±0 | 0±0 | 0±0 | 0±0 | 0±0 |
| | **Meta-Poison** | 42.5±2.8 | 30.7±2.3 | 28.1±2.5 | 19.4±2.1 | 5.2±1.7 | 3.9±1.2 |
| | **Grad-Match** | 72.9±3.4 [2] | 63.4±3.8 | 35.4±3.1 | 26.8±3.4 | 10.3±2.6 | 6.4±2.8 |
| | **SAPA+Grad-Match** | **80.1±3.5** | **70.8±2.7** | **48.4±2.8** | **31.5±3.1** | **16.7±2.4** | **11.2±1.5** |
| **CIFAR100** | **Bullseye** | 2.1±0.3 | 0±0 | 2.6±0.7 | 1.2±0.2 | 0.5±0.1 | 0.1±0.1 |
| | **Poison-Frog** | 1.0±0.8 | 0±0 | 1.1±0.4 | 0±0 | 0.2±0.1 | 0±0 |
| | **Meta-Poison** | 50.3±2.6 | 24.1±2.5 | 43.2±2.8 | 22.7±1.7 | 4.5±1.2 | 3.1±0.8 |
| | **Grad-Match** | 90.2±3.1 | 53.6±2.5 | 62.1±3.5 | 33.4±4.4 | 11.3±2.5 | 7.4±1.9 |
| | **SAPA+Grad-Match** | **91.6±1.5** | **74.9±2.2** | **86.8±3.9** | **52.6±3.6** | **31.6±2.4** | **12.1±1.2** |

## 5.2 PERFORMANCE OF SAPA IN BACKDOOR ATTACKS

**Experiment setup**. In this subsection, we study the effectiveness of SAPA in backdoor attacks. In particular, we focus on the "hidden-trigger" setting (Saha et al., 2020; Souri et al., 2022) where the attackers can only add imperceptible perturbations to the clean training samples. For backdoor attacks, the adversarial goal is to cause the samples from a victim class $y^{vic}$ to be wrongly classified

---

[2]Results for baseline Grad-Match is lower than reported in the original paper, because the original paper only trains 40 epochs. Detailed analysis on the impact of epochs can be found in Appendix D

as a designated class $y^{obj}$ by inserting triggers. Besides, we follow the setting in (Souri et al., 2022) that the attacker adds imperceptible perturbations to samples in class $y^{obj}$. Therefore, in our study, we randomly choose the two different classes $y^{vic}, y^{obj}$ for poisoning sample generation. In this evaluation, we constrain that the attacker can only perturb 1% of the whole training set and the re-training process resembles our settings in Section 5.1. All experiments are repeated for 50 times and we report the average success rate. More implementation details are in Appendix.C.2

**Baselines**. In the experiment, our proposed method SAPA is incorporated to Sleeper Agent (Souri et al., 2022) which also leverages Gradient Match (Geiping et al., 2020) to achieve the adversarial goal in backdoor attacks. We also show the results of the method, Hidden-Trigger Backdoor (Saha et al., 2020), which optimizes the poison over images with triggers to preserve information of triggers, and the Clean-Label Backdoor method (Turner et al., 2019) leverages calculating adversarial examples (Goodfellow et al., 2014) to train a backdoored model.

**Performance comparison**. Our results are shown in Table 2, where we report the "Success Rate", which is the ratio of samples (with triggers) in $y^{vic}$ that are classified as $y^{obj}$ by the poisoned model. From the result, our method outperforms all baselines under all settings. Specifically, Hidden-trigger (Saha et al., 2020) and Clean-label (Turner et al., 2019) suffer from low ineffectiveness, as they are either designed for transfer learning or require control over the training process. Compared to these methods, SAPA shows effectiveness for different model architectures and perturbation budgets. In comparison with Sleeper Agent (Souri et al., 2022), which is also based on Gradient Matching (Geiping et al., 2020), our method can also have a clear improvement. Especially, under the perturbation budget 8/255 and CIFAR100 dataset, SAPA can obviously outperform Sleeper Agent.

Table 2: Success Rate in Backdoor Attacks on CIFAR10 and CIFAR100

|  |  | ResNet18 | | ResNet50 | | VGG11 | |
|---|---|---|---|---|---|---|---|
|  |  | 16/255 | 8/255 | 16/255 | 8/255 | 16/255 | 8/255 |
| CIFAR10 | Hidden-trigger | 3.5±1.2 | 1.3±0.4 | 3.2±0.9 | 1.3±0.3 | 5±1.4 | 1.8±0.7 |
|  | Clean-label | 2.7±1.1 [3] | 0.9±0.7 | 2.6±0.7 | 0.9±0.2 | 4.7±1.5 | 1.1±0.9 |
|  | Sleeper Agent | 90.9±2.2 | 31.5±4.2 | 94.1±2.7 | 21.2±4.3 | 85.8±3.4 | 26.7±3.9 |
|  | SAPA+Sleeper Agent | **98.1±2.6** | **49.3±3.7** | **98.4±4.9** | **33.2±3.1** | **94.3±2.6** | **35.5±2.8** |
| CIFAR100 | Hidden-trigger | 2.1±0.9 | 1.3±0.4 | 1.7±0.3 | 0.8±0.1 | 3.4±1.2 | 1.2±0.6 |
|  | Clean-label | 1.5±0.7 | 0.9±0.2 | 1.2±0.4 | 0.4±0.1 | 2.6±0.3 | 0.8±0.2 |
|  | Sleeper Agent | 58.3±3.9 | 26.7±4.3 | 47.2±4.5 | 18.5±3.7 | 41.6±3.2 | 12.9±2.6 |
|  | SAPA+Sleeper Agent | **72.4±3.6** | **41.8±3.2** | **63.9±3.5** | **31.4±3.2** | **67.7±2.7** | **30.3±2.3** |

## 5.3 PERFORMANCE OF SAPA IN UN-TARGETED ATTACKS

**Experiment Setup**. The goal of un-targeted attacks is to degrade the models' test accuracy. We follow the line of existing works (Huang et al., 2021; Fowl et al., 2021; Ren et al., 2022) (which are also called "un-learnable examples") to perturb a large portion of training samples (50%, 80%, 100%) in CIFAR10 and CIFAR100 datasets, in order to protect the data against being learned by DNNs. In our experiment, we limit the perturbation budget to 8/255. We first generate poisoning samples targeting on ResNet18. Then we re-train the victim models under ResNet18 and ResNet50 following existing works (Huang et al., 2020; Fowl et al., 2021). The training procedure for each model also resembles the settings in Section 5.1, and we repeat experiments for each setting 5 times and report the average performance. More details of the implementation are in Appendix C.3.

**Baselines**. We compare SAPA with representative "un-learnable" methods, such as Error-Min (Huang et al., 2020), Error-Max (Fowl et al., 2021), Separable Perturbation (Yu et al., 2022), and Autoregressive Perturbation (Sandoval-Segura et al., 2022). We also report the clean performance which refers to the accuracy of models without poisoning attack. Notably, our proposed method SAPA can be incorporated into either Error-Min or Error-Max, so we denote our method as "Error-Min+SAPA" and "Error-Max + SAPA" respectively. We provide more details of the algorithm of "Error-Max+SAPA" in Appendix C.

**Performance comparison.** In Table 3, we report the accuracy of the re-trained model on the clean test dataset of CIFAR10 and CIFAR100, so a lower value indicates better attack performance. From the result SAPA can improve the poisoning effect for both Error-Min and Error-Max under all settings. For example, when incorporating SAPA to Error-Min, Error-Min+SAPA has a clear advantage, by reducing the accuracy to around 10% when the poisoning ratio is 100% in the CIFAR10 dataset.

---

[3]Hidden trigger and Clean-label also have lower success rates than in their original papers. This is because they are originally proposed for fine-tuning settings (by fixing the early layers), while we focus on end-to-end setting in this work. Our results are consistent with results in work (Souri et al., 2022).

In other settings, Error-Min+SAPA can also manage to achieve a 2-4% accuracy reduction compared to Error-Min. Similarly, Error-Max+SAPA is also demonstrated to have a consistent improvement over Error-Max. In addition, incorporating our strategy can boost Error-Max and Error-Min to achieve comparable performance with Autoregressive Perturbation which specifically targets on the vulnerability of CNNs.

Table 3: Test Accuracy of Models Trained on Poisoned Datasets via Un-targeted Attacks.

| | CIFAR10 | | | | | | CIFAR100 | | | | | |
|---|---|---|---|---|---|---|---|---|---|---|---|---|
| | ResNet18 | | | ResNet50 | | | ResNet18 | | | ResNet50 | | |
| | 100% | 80% | 50% | 100% | 80% | 50% | 100% | 80% | 50% | 100% | 80% | 50% |
| Clean* | 94.8 | 94.8 | 94.8 | 95.0 | 95.0 | 95.0 | 74.8 | 74.8 | 74.8 | 75.2 | 75.2 | 75.2 |
| Separable. | 13.5 | 86.3 | 92.9 | 14.9 | 88.1 | 93.2 | 9.1 | 57.1 | 66.2 | 8.4 | 60.8 | 66.7 |
| Autoregressive. | 11.8 | **82.3** | **89.8** | **10.1** | **83.6** | **90.3** | 4.2 | **51.6** | **64.7** | **4.3** | **53.5** | **66.1** |
| Error-Max | 11.9 | 88.2 | 92.2 | 12.8 | 90.1 | 93.9 | 4.8 | 57.3 | 66.9 | 5.6 | 58.3 | 68.1 |
| Error-Max+SAPA | **9.6** | 84.6 | 90.1 | 10.9 | 85.7 | 91.3 | **4.1** | 55.1 | 64.8 | 4.9 | 56.8 | 66.9 |
| Error-Min | 21.2 | 87.1 | 93.4 | 18.9 | 89.5 | 94.5 | 11.2 | 56.9 | 67.7 | 10.8 | 60.5 | 70.3 |
| Error-Min+SAPA | 10.9 | 83.7 | 90.0 | 10.3 | 85.2 | 91.8 | 8.7 | 53.1 | 65.3 | 9.5 | 57.9 | 67.6 |

## 5.4 ROBUSTNESS OF RE-TRAINING VARIANTS

We provide ablation studies on the effectiveness of SAPA under various re-training settings including different re-training algorithms, re-training schedules and model architectures. We also study the robustness of SAPA against adversarial training (Madry et al., 2017). Notably, we only focus on the CIFAR10 dataset and ResNet18 (except for the studies on various architectures). For targeted and backdoor attacks, we specify the perturbation budget to $16/255$ with poisoning ratio $\epsilon = 1\%$. For un-targeted attacks, we specify the perturbation budget to be $8/255$ with poisoning ratio $\epsilon = 100\%$.

**Various Re-training Algorithms**. There are studies (Schwarzschild et al., 2021; Ren et al., 2022) demonstrating that many poisoning attacks can lose efficacy when faced with training algorithms beyond Empirical Risk Minimization (ERM). Therefore, we provide additional experiments to test the performance of SAPA under various re-training algorithms. We mainly consider the algorithms including Cut-Out (DeVries & Taylor, 2017), Mix-Up (Zhang et al., 2017), and Sharpness-aware Minimization (SAM) (Foret et al., 2020), which is proposed to minimize model sharpness to improve model generalization. Two optimization methods—SGD and ADAM are also included. In Table 4, we compare SAPA with its backbone attack as baselines for each type of poisoning attack, and we use the same evaluation metric as previous subsections. From this table, we can see that our method remains outperforming the baseline attacks. Notably, among these re-training algorithms, Mix-Up shows an outstanding ability to reduce the poisoning effect for all attack methods that we studied. It may be because Mix-Up is a data augmentation strategy which drastically manipulates the training data distribution, which can weaken the poisoning effect of the injected poisoning samples.

Table 4: SAPA vs Strongest Baselines under Re-training Scheme Variation.

| | Un-targeted($\downarrow$) | | Targeted($\uparrow$) | | Backdoor($\uparrow$) | |
|---|---|---|---|---|---|---|
| | Error-min | SAPA | GradMatch | SAPA | SleeperAgent | SAPA |
| SGD | 21.2 | 10.9 **(-10.3)** | 73.1 | 80.1 **(+6.8)** | 91.8 | 97.1 **(+5.3)** |
| ADAM | 19.7 | 10.4 **(-9.3)** | 80.2 | 85.4 **(+5.2)** | 92.3 | 97.7 **(+5.4)** |
| Cut-Out | 22.6 | 11.2 **(-11.4)** | 82.4 | 90.3 **(+7.9)** | 97.8 | 100.0 **(+2.2)** |
| Mix-Up | 40.8 | 36.7 **(-4.1)** | 58.4 | 65.5 **(+7.1)** | 69.7 | 76.5 **(+6.8)** |
| SAM | 28.9 | 11.3 **(-17.6)** | 74.3 | 80.7 **(+6.4)** | 79.9 | 85.3 **(+5.4)** |
| | Un-targeted($\downarrow$) | | Targeted($\uparrow$) | | Backdoor($\uparrow$) | |
| | Error-min | SAPA | GradMatch | SAPA | SleeperAgent | SAPA |
| ResNet18 | 21.2 | 10.9 **(-10.3)** | 73.1 | 80.1 **(+7.0)** | 91.8 | 97.1 **(+5.3)** |
| MobileNetV2 | 21.5 | 11.9 **(-9.6)** | 68.5 | 75.2 **(+6.7)** | 30.2 | 37.6 **(+7.4)** |
| VGG11 | 35.3 | 20.2 **(-15.1)** | 42.9 | 47.6 **(+4.7)** | 31.9 | 37.4 **(+5.5)** |
| ViT | 40.6 | 36.5 **(-4.1)** | 36.2 | 41.5 **(+5.3)** | 24.7 | 26.3 **(+1.6)** |

**Various model architectures**. We also test different model architectures after the generation of poisoned data for all three types of attacks. In detail, we test on MobileNetV2 (Sandler et al., 2018), VGG11 (Simonyan & Zisserman, 2014) and pretrained Vision Transformer(ViT, (Dosovitskiy et al., 2020)). Results in Table 4 conclude that SAPA consistently improves the baseline methods, and shows a better ability to adapt to unknown model architectures during re-training.

**Various Re-Training Schedules**. There are also studies (Schwarzschild et al., 2021; Huang et al., 2020) suggesting that a different re-training epoch number or schedule can significantly break the poisoning effect. Therefore, we provide additional experiments when: (1) the model is trained for

500 epochs and the learning rate is updated by "steps" similar to previous studies, and (2) the re-training learning rate is updated "cyclicly". In Figure 1, we plot the curve of the poisoning effect of SAPA and baselines in the backdoor and un-targeted attacks. Note that we exclude targeted attacks because the poisoning effect is discrete in one model. From these figures, we can find that SAPA can stably converge to the point with a strong poisoning effect that consistently outperforms baselines.

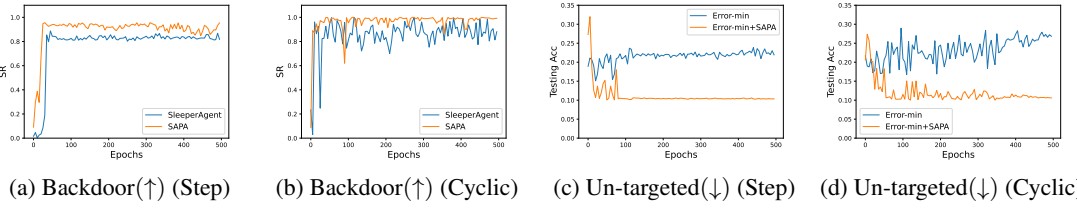

(a) Backdoor(↑) (Step)  (b) Backdoor(↑) (Cyclic)  (c) Un-targeted(↓) (Step)  (d) Un-targeted(↓) (Cyclic)

Figure 1: The Curve of Poisoning Effect for Various Re-training Schedules

**Adversarial Training**. In reality, these attacks can be faced to defense strategies. In this part, take untargeted attack as an example, we conduct a case study to evaluate the resistance of SAPA against Adversarial Training (Madry et al., 2017), which is a strong defense for un-targeted attacks (Tao et al., 2021). We test on SAPA as well as all baselines in un-targeted attack. From the results, we note that adversarial training can

Table 5: Adv. Train.

|  | ERM | Adv Train. |
| --- | --- | --- |
| Separable | 11.8 | 75.2 |
| Autoregressive | 13.5 | 78.7 |
| Error-min | 21.2 | 82.0 |
| Error-min + SAPA | 10.9 | 73.2 |
| Error-max | 11.9 | 80.3 |
| Error-max + SAPA | 9.6 | 72.5 |

significantly increase the test accuracy to defend against all attack methods. However, SAPA can can still outperform other attacks. Notably, although Adv. Train. is usually considered as a strong defense to untargeted attacks, it can naturally reduce the accuracy on CIFAR10 from $95\%$ to $85\%$ without any poisoning attacks, based on extensive previous studies (Tsipras et al., 2018).

### 5.5 EFFICIENCY VS EFFECTIVENESS TRADE-OFF

Previous works (Huang et al., 2020) also leverage the *Ensemble* and *Re-initialization* (E&R) technique, to take various model architectures and initializations into consideration to handle re-training uncertainty. In this part, we compare the efficiency and effectiveness trade-off between SAPA and E&R, when they are incorporated to existing attacks, such as Gradient Matching and Sleeper Agent. In Figure 2, we report the attack successful rate (ASR) and computing time of SAPA and E&R with different options ($K, R$ denote the number of ensembled model architectures and initializations respectively). Specifically, SAPA has higher ASR when $K$ and $R$ are small, and can still achieve comparable success rates when $K$ and $R$ are increasing. However, the running time grows dramatically for large $K$ and $R$ making them much less efficient than SAPA. This result shows SAPA demonstrate much better efficiency and effectiveness trade-off compared with E&R. Notably, we exclude the result for untargeted attacks, as it generates poisoning samples for the whole training set, which makes E&R extremely inefficient.

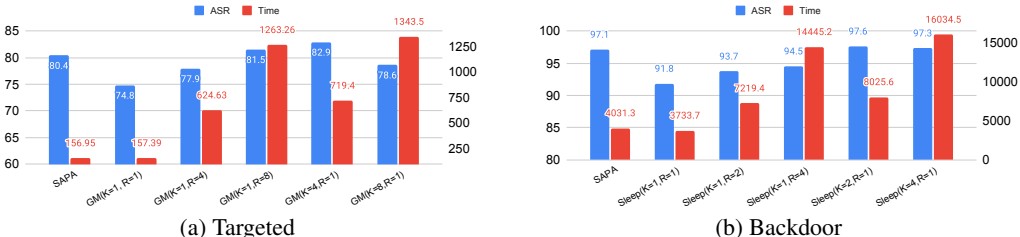

(a) Targeted  (b) Backdoor

Figure 2: ASR and running time for targeted attack (GM) and backdoor attack (Sleep.).

## 6 CONCLUSION AND LIMITATION

In this paper, we introduce a novel and versatile framework for data poisoning attacks that takes into account the model landscape sharpness. We apply this strategy to various types of poisoning attacks, including un-targeted, targeted, and backdoor attacks. Our experimental results demonstrate the superiority of our method compared to existing approaches. However, in this paper, our method focuses on image classification. Therefore, we will leave the relative studies, such as the poisoning attacks in self-supervised learning settings, and poisoning attacks in other domains such as texts and graphs for future investigation.

## 7 ACKNOWLEDGEMENT

This research is supported by the National Science Foundation (NSF) under grant numbers CNS 2246050, IIS1845081, IIS2212032, IIS2212144, IOS2107215, DUE 2234015, DRL 2025244 and IOS2035472, the Army Research Office (ARO) under grant number W911NF-21-1-0198, the Home Depot, Cisco Systems Inc, Amazon Faculty Award, Johnson&Johnson, JP Morgan Faculty Award and SNAP.

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

## A  BOARDER IMPACT

Our research unveils a powerful attacking framework that has the potential to compromise security-critical systems in a stealthy manner. As machine learning models, particularly large models that require extensive training datasets, become increasingly prevalent and assume significant roles in various domains, the importance of clean training data cannot be overstated. It is imperative to prioritize the quality of data to ensure the success and reliability of these models. By shedding light on the potential dangers associated with this threat model, our study aims to raise awareness about the importance of data security. We hope that our findings will serve as a catalyst for the development of stronger defenses against data poisoning attacks. Safeguarding against these threats requires a proactive approach and increased vigilance to protect the integrity and robustness of machine learning systems.

## B  DISCUSSION OF EXISTING DATA POISONING ATTACKS.

### B.1  THREAT MODEL AND OBJECTIVES

In this section, we introduce the threat model, adversarial objective, and victim set for each type of data poisoning attack.

**Un-targeted Attacks**. In un-targeted attacks (Steinhardt et al., 2017), the attacker aims to cause the trained model with an overall low test accuracy. The attackers are assumed to have access to the training data and can perturb part (Steinhardt et al., 2017) or the whole training data (Fowl et al., 2021; Huang et al., 2021). However, because the attacker usually does not have knowledge of test distribution and the training process of the victim model, most works (Steinhardt et al., 2017; Fowl et al., 2021; Huang et al., 2021) define the adversarial objective as the following to maximize the model error on the clean training set $D_{tr}$:

$$Q_{ut}(\theta^*, D_{tr}) = -L(\theta^*; D_{tr}) \tag{14}$$

**Targeted attacks**. In targeted attacks (Shafahi et al., 2018), the attacker aims to cause the trained model to misclassify a specified test sample or a subset of test samples. For example, they are targeting on a victim person and have knowledge of $k$ $(k \geq 1)$ photographs of this person[4] $\{(x_i^{vic}, y^{vic})\}_{i=1}^k$. They aim to cause the model to misclassify the photos of this person while preserving the overall accuracy of the rest. The attackers are only allowed to perturb a small part of the training data and have no knowledge of the victim model including the initialization and training algorithm. Therefore, they define the victim set $D_T = \{(x_i^{vic}, y^{vic})\}_{i=1}^k$ and the adversarial objective as:

$$Q_t(\theta^*, D_T) = \sum_{(x_i, y_i) \in D_T} l(f(x_i; \theta^*), y^{obj}), \tag{15}$$

where $y^{obj}$ is a designated class different from $y^{vic}$.

**Backdoor attacks**. In backdoor attacks (Chen et al., 2017; Souri et al., 2022), the attacker aims to take control of the model prediction by injecting samples with "triggers". In particular, if there is a trigger, such as a patch $p$, present in an image, the poisoned model will predict this sample to a specified class $y^{obj}$. Otherwise, the poisoned model will make a correct prediction. Similar to the targeted attack, attackers are only allowed to perturb a small part of training data and have no control of the training process of the victim model. In backdoor attacks, most works target on samples from a specific victim class $y = y^{vic}$ and define the victim set as $D_T = \{(x, y) \in D_{tr} | y = y^{vic}\}$. During the attack, they aim to solve the adversarial objective:

$$Q_b(\theta^*, D_T) = \sum_{(x_i, y_i) \in D_T} l(f(x_i \oplus p; \theta^*), y^{obj}) \tag{16}$$

where $x \oplus p$ denotes the process that $p$ is attached to a clean image $x$. In this way, the poisoned model is likely to predict the samples with triggers to be class $y^{obj}$.

---

[4]We assume that the samples of the victim are from the same class $y^{vic}$, following most existing works (Shafahi et al., 2018).

## B.2 ALGORITHMS

In this section, we discuss the details of existing data poisoning attacks.

**Targeted attacks.** Targeted attacks insert poisons into the clean training aiming at misclassifying targets(samples or classes) as adversarial labels. Fundamentally, targeted attacks can be formulated as a bi-level optimization problem in Eq.15, which can be challenging especially for DNN because the inner problem has multiple minima. Some existing methods avoid solving it directly and apply heuristic approaches such as Feature Collision Attacks(i.e. Bullseye(Aghakhani et al., 2021), poison-frog(Shafahi et al., 2018)) which manipulate the representation of victims to mislead models classify them as adversarial labels. These methods are well-suited for the transfer learning setting(Goldblum et al., 2022), where a model is pre-trained on clean data and fine-tuned on a smaller poisoned dataset. However, this brings a natural drawback in that its poisoning effect is only kept for one surrogate model(the pre-trained model), thus these methods can hardly work for the retrain-from-scratch scenario as the retrained model can be very different from the surrogate model. Another line of work tries to handle the bi-level optimization problem directly. For linear or convex models, the inner problem typically exhibits a single minimum. As a result, several methods such as those proposed by(Biggio et al., 2012; Xiao et al., 2015) have been developed to successfully achieve the malicious objective. Theoretical analyses, such as the work by(Mei & Zhu, 2015), have provided guarantees in these cases. However, these methods become ineffective for DNN. (Muñoz-González et al., 2017) applies a method called "back-gradient descent" in which the inner problem is approximately solved using several steps of gradient descent, and the gradient-descent for the outer is conducted by back-propagating through the inner minimization routine. This method is both time and memory-consuming, thus impractical for models with multiple layers. MetaPoison(Huang et al., 2020) draws inspiration from (Franceschi et al., 2018; Shaban et al., 2019) and unrolling the training pipeline of inner. They also apply "ensembling" and "network re-initialization" to avoid overfitting to one single surrogate model and try to preserve the poisoning effect during re-training. However, the success of this unrolling still requires single minima assumption(Liu et al., 2020), leading to less effectiveness as shown in the empirical results. Grad-Match(Geiping et al., 2020) leverages a "gradient alignment" technique to solve the problem in Eq.15 and applies "poison re-initialization" to select better poisons, but there still exists space for improvement as the formulation does not capture the nature of inner multiple minima well(Sow et al., 2022). This is the reason why we rethink the formulation of targeted attacks and design a new objective in Eq.5.

**Backdoor attacks.** Backdoor attacks aim at inserting triggers into training samples and testing samples, such that triggered inputs will be misclassified. Note that there exist many types of backdoor attacks(Wu et al., 2022) with regard to factors of backdoor, such as attacker's capability(training controllable or not), characteristics of triggers(visibility, etc). In Eq.16, we focus on so-called *hidden-trigger clean-label* backdoor attacks, meaning attackers insert poisoned samples rather than triggers into the training data, and testing inputs attached with a trigger will be misclassified. Hidden-trigger(Saha et al., 2020) generates poisoned images carrying information about triggered source images and confuses the model during fine-tuning. Clean-label(Turner et al., 2019) leverages adversarial perturbations to cause a successful backdoored effect. However, both methods can hardly succeed in the retrain-from-scratch scenario, as Hidden-trigger is designed for transferring learning and Clean-label needs to control the retraining process. Sleeper Agent(Souri et al., 2022) incorporates the gradient matching technique to solve the problem in Eq.16, and considers "model restarts" to adaptively update the model during the generation of poisons, but it can only cover a few models and limits its effectiveness. Our SAPA utilizes the sharpness objective to capture the nature of multiple-inner-minima, and is shown to better preserve the poisoning effect during retraining. It is worth noting that there exists a line of work that consider a different scenario from end-to-end scenario discussed in this paper. Some of these attacks need additional assumptions that the attacker can control the re-training process such as WaNet (Nguyen & Tran, 2021) and LiRA (Doan et al., 2021); some conduct attacks without involving any models and require noticeable triggers, such as BadNet (Gu et al., 2019). Therefore, this line of work is out of the main scope of our paper.

**Un-targeted attacks.** Un-targeted attacks insert poisons into the clean training and the goal is to degrade the overall accuracy over the clean testing. Generally, it can be formulated as a bi-level optimization problem in Eq.14. Early works(Biggio et al., 2012; Steinhardt et al., 2017; Koh et al., 2022) primarily focused on simple models with desirable properties like linearity and convexity. In these cases, the inner problem typically possesses a unique solution, and these methods demonstrate effec-

tive performance. Theoretical analysis(Steinhardt et al., 2017; Suya et al., 2021) have been carried out to establish the feasibility of untargeted attacks on such simple models. Nevertheless, the presence of non-convexity and multiple minima within the inner problem poses significant challenges for untargeted attacks on DNN models. (Muñoz-González et al., 2017) applies "back-gradient descent", which is expensive and infeasible for complex models. (Lu et al., 2022) borrows ideas from Stackelberg games and applies so-called "total gradient descent ascent" techniques to generate poison samples one by one. However, they can hardly preserve poisoning effects after retraining and have a subtle influence on clean testing. CLPA(Zhao & Lao, 2022) takes advantage of 'naturally poisoned data' which refers to the wrongly predicted samples by the clean model and generates them using GAN. Nevertheless, this method can only be applied for transferring learning as it relies on the clean model and suffers from the multiple-inner-minima problem. MetaPoison(Huang et al., 2020) tries to handle the multiple-inner-minima problem using ensembling and re-initialization, but the performance is not very satisfactory. Error-max(Fowl et al., 2021) solves the problem in Eq.14 by leveraging adversarial examples, and Error-min(Huang et al., 2021) inserts perturbations to build a relationship between labels and perturbations. These two methods only involve one surrogate model and can be improved through our proposed method. There also exists another line of work that generate perturbations without access to model or datasets, such as Synthetic Perturbations(Yu et al., 2022) and Autoregressive Perturbations(Sandoval-Segura et al., 2022). Though they do not solve the optimization problem directly and do not have the multiple-inner-minima problem, their application may be limited to specific settings(such as un-targeted attacks and CNN models), and not as general as ours.

## C   IMPLEMENTATION DETAILS OF ALL SAPA METHODS

In this subsection, we provide details of the implementation of SAPA methods.

### C.1   TARGETED ATTACKS

As introduced in Section.4.1, we solve the optimzation problem in Eq.8 using Gradient Matching. To be more specific, we randomly choose one or a few target samples from a victim class(in testing) as the victim set $D_T$. At the same time, we randomly choose a percentage of samples $D_p$ from the training to be modified in the algorithm, and the percentage is referred to as *poisoning ratio*. The attacking goal is to make the model trained on $D_{tr} + D_p$ misclassify samples in $D_T$ as a pre-specified adversarial class(different from the true class). To ensure the imperceptibility of poisons, we rigorously control the *poisoning ratio* and *perturbation budget*($L_\infty$ norm of the perturbation) and in our experiments, we consider the ratio of $1\%, 0.2\%$ and budget of $16/255, 8/255, 4/255$.

Given a pre-trained clean model, we first randomly initialize the perturbation for $D_p$, and during each optimization step, we compute the gradient of the sharpness-aware objective using Eq.12 and do one step gradient descent to minimize the similarity loss defined in Eq.10. The detailed algorithm is shown in Algorithm.2.

After the generation of poisons, we retrain the model from scratch via SGD for 160 epochs with an initial learning of 0.1 and decay by 0.1 at epochs 80,120. After training, we evaluate the prediction on targets $D_T$. For the single-target case, only predicting the target as the adversarial label is considered a success, and we sample 10 random poison-target cases as one trial, for which the average success rate over 10 cases is referred to as the success rate for this trial. For each setting(poisoning ratio and perturbation budget), we do 50 trials and report the average success rate. For the multi-target case, we randomly choose a few targets, 4 or 8 in our experiments, and the average accuracy of predicting targets as adversarial labels is referred to as the success rate of one experiment. We also repeat experiments 50 times and report the average success rate.

### C.2   BACKDOOR ATTACKS

Similar to the targeted attack, we leverage the Gradient Matching method. To be more specific, we randomly choose a victim class and an objective class. Then we randomly sample the victim set $D_T$ from the victim class and select $D_p$ from the objective class to be modified in the algorithm. Note that both $D_T$ and $D_p$ are sampled from the training data. The attacking goal is to make the model retrained on $D_{tr} + D_p$ misclassify images from the victim class(in testing) which are attached with

---

**Algorithm 2:** SAPA in Targeted Attacks(Grad-Match)

---

**Input** : Pre-trained model $f(\cdot; \theta^*)$ on the clean training set $D_{tr}$; a victim set $D_T$; optimization
      step $M$.
Randomly initialize the poisoning samples $D_p$
Compute the gradient $\nabla_\theta Q^S(\theta^*; D_T)$ using Eq.12
**for** $m = 1, ..., M$ **do**
  | Update $D_p$ in Eq.10 with one-step gradient descent
**end**
**Output:** Return $D_p$

---

a pre-specified trigger as the objective class. We also restrict the *poisoning ratio* to be 1% and the perturbation budget is bounded by $16/255, 8/255$.

Given a pre-trained clean model, we first randomly initialize the perturbations for $D_p$, and during each optimization step, we compute the gradient in Eq.12, but different from targeted attacks, in Eq.12 $D_T$ are attached with the trigger which will be used to backdoor images in the testing. Then we do one-step gradient descent to minimize the similarity loss defined in Eq.10, after optimizing for $R$ steps, we update the model $f$ on poisoned training data $D_{tr} + D_p$, and then continue optimizing perturbations on the updated model. The detailed algorithm is shown in Algorithm.3.

After the generation, we retrain the model from scratch via SGD for 160 epochs with an initial learning of 0.1 and decay by 0.1 at epochs 80,120. After that, we evaluate the prediction of triggered images from the entire victim class in the testing. The average accuracy of predicting triggered images as the adversarial label is referred to as the success rate. We repeat experiments 50 times and report the average accuracy as the final results. We conduct experiments on multiple datasets including CIFAR10, CIFAR100 and SVHN, along with multiple victim models including ResNet18, ResNet50 and VGG11.

---

**Algorithm 3:** SAPA in Backdoor Attacks

---

**Input** : Pre-trained model $f(\cdot; \theta^*)$ on the clean training set $D_{tr}$; a victim set $D_T$; retraining
      factor $R$; optimization step $M$.
Randomly initialize the poisoning samples $D_p$
**for** $m = 0, ..., M - 1$ **do**
  | **if** $m \bmod \lfloor M/(R+1) \rfloor = 0$ *and* $m \neq M$ **then**
  |   | Update $\theta^*$ on poisoned training $D_{tr} + D_p$
  |   | Find the worst-case direction $\hat{v}$ using Eq.11
  |   | Approximate $\nabla_\theta^S Q(\theta^*; D_T)$ by $\nabla_\theta Q(\theta^*; D_T)|_{\theta=\theta^*+\hat{v}}$.
  | **end**
  | Update $D_p$ with one step gradient descent
**end**
**Output:** Poisoning set $D_p$

---

### C.3 UN-TARGETED ATTACKS

We implement SAPA based on two existing methods: Error-min(Huang et al., 2021) and Error-max(Fowl et al., 2021).

**Error-min+SAPA**. We have discussed this method in Section 4.3, and the detailed algorithm is shown in Algorithm.1. In our experiments, we set $T = 20$, $M = 10$, $\alpha = \epsilon/10$ and $E = 100$.

**Error-max+SAPA**. Different from Error-min, Error-max is used to solve the optimization as follows:

$$\max_{\{\delta_i\}_{i=1,...,n}} \sum_{(x_i, y_i) \in D_{tr}} l\big(f(x_i + \delta_i; \theta^*), y_i\big)$$

where $\theta^*$ denote a model trained on clean data and is fixed during poison generation. Thus SAPA+Error-max focuses on the sharpness-aware objective as follows:

$$\max_{\{\delta_i\}_{i=1,\ldots,n}} \left[ \max_{||v|| \leq \rho} \sum_{(x_i,y_i) \in D_{tr}} l\big(f(x_i + \delta_i; \theta^* + v), y_i\big) \right]$$

Similar to Error-Min+SAPA, we also solve this optimization problem using gradient descent. As shown in Algorithm 4, given the clean pre-trained model $f(\cdot, \theta)$, we first find the worst direction $\hat{v}$ to maximize $\max_v \sum_{(x_i,y_i) \in D_{tr}} l\big(f(x_i + \delta_i; \theta^* + v), y_i\big)$ and then update $\delta_i$ fixing $\theta^* + \hat{v}$.

After generating perturbations via either SAPA+Error-min or SAPA+Error-max, we retrain the model on poisoned training via SGD for 160 epochs with an initial learning of 0.1 and decay by 0.1 at epochs 80,120. Then we evaluate the prediction of clean testing and report the average accuracy. Our experiment is conducted on multiple datasets including CIFAR10, CIFAR100, SVHN, along with multiple victim models including ResNet18 and ResNet50.

---

**Algorithm 4:** Error-max+SAPA

---

**Input** : Network $f(\cdot; \theta)$; clean training set $\{(x_i, y_i)\}_{i=1}^n$; perturbation bound $\epsilon$; PGD step $T$; pre-train steps $R$

**Output:** Sample-wise perturbation $D_p = \{\delta_i\}_{i=1}^n$

**for** *r in* $1, ..., R$ **do**
  | Update $\theta$ via minimizing $L(\theta; D_{tr})$
**end**

Randomly initialize perturbation $D_p$

Fix $\theta, D_p$, find the worst direction $\hat{v}$ to maximize $\sum_{(x_i,y_i) \in D_{tr}} l\big(f(x_i + \delta_i; \theta^* + v), y_i\big)$

**for** $t = 1, ..., T$ *PGD steps* **do**
  | Update $D_p$ via gradient ascent fixing $\theta + \hat{v}$
**end**

---

# D    ADDITIONAL EXPERIMENTS

In this section, we provide additional experiments to further illustrate the effectiveness of SAPA.

## D.1    ADDITIONAL DATASET

We test on the additional dataset, SVHN, to further illustrate the advantage of our method. All the experiments are conducted following the same procedure as in Section.5, while some details are different. For un-targeted attacks, we repeat all experiments on SVHN. For targeted attacks, we conduct on the "Single-Victim" case and omit baselines Poison-Frog and Bulleyes because they do not perform well under the train-from-scratch setting. For backdoor attacks, we also omit baselines Clean-label and Hidden-trigger which do not perform well.

**Performance comparison.** Results of un-targeted attacks are shown in Table.6. Similar to results on CIFAR10 and CIFAR100, our method outperforms nearly all baselines under all settings. Results of targeted attacks are shown in Table.7. Our method outperforms all baselines significantly especially for smaller budget sizes and poison ratios. Results of backdoor attacks are shown in Table.8. Our method has better performance than baselines under all settings.

Table 6: Test Accuracy of Models Trained on Poisoned Datasets via Un-targeted Attacks.

|  | SVHN | | | | | |
|---|---|---|---|---|---|---|
|  | ResNet18 | | | ResNet50 | | |
|  | 100% | 80% | 50% | 100% | 80% | 50% |
| Clean$^*$ | 96.0 | 96.0 | 96.0 | 95.9 | 95.9 | 95.9 |
| Separable | 8.3 | 92.5 | 94.3 | 7.8 | 90.9 | 93.1 |
| Autoregressive. | 7.2 | 89.5 | 92.5 | 7.1 | 89.1 | 91.2 |
| Error-Max | 5.3 | 92.9 | 93.4 | 5.7 | 92.5 | 93.7 |
| Error-Max+SAPA | 4.7 | 92.3 | 92.6 | 5.1 | 91.8 | 92.1 |
| Error-Min | 13.8 | 92.8 | 95.3 | 13.7 | 91.3 | 94.2 |
| Error-Min+SAPA | 10.2 | 91.7 | 93.1 | 11.4 | 90.4 | 91.8 |

Table 7: Success Rate under the "Single-Victim" Setting in Targeted Attacks.

|  |  | 16/255 | | 8/255 | | 4/255 | |
|---|---|---|---|---|---|---|---|
|  |  | $\epsilon = 1\%$ | $\epsilon = 0.2\%$ | $\epsilon = 1\%$ | $\epsilon = 0.2\%$ | $\epsilon = 1\%$ | $\epsilon = 0.2\%$ |
|  | Meta-Poison | 52.1 | 28.7 | 35.3 | 21.9 | 23.8 | 11.2 |
| SVHN | Grad-Match | 62.6 | 37.4 | 44.2 | 25.7 | 31.3 | 17.4 |
|  | SAPA | **71.2** | **48.5** | **55.7** | **38.2** | **42.1** | **23.3** |

Table 8: Success Rate in Backdoor Attacks on SVHN

|  | ResNet18 | | ResNet50 | | VGG11 | |
|---|---|---|---|---|---|---|
|  | 16/255 | 8/255 | 16/255 | 8/255 | 16/255 | 8/255 |
| Sleeper Agent | 92.3 | 42.6 | 91.6 | 34.8 | 86.2 | 38.1 |
| SAPA-backdoor | **97.5** | **58.3** | **95.4** | **44.1** | **91.4** | **43.7** |

## D.2    MULTI-TARGET TARGETED ATTACKS

We also perform tests on poisoning multiple targets simultaneously. We conduct experiments on model ResNet18 and two datasets Cifar10 and Cifar100. We only compare with the most powerful baseline Grad-Match under multiple settings, including different number of victim targets(4,8), perturbation size(16/255, 8/255 and 4/255) and poisoning rate(1% and 0.25%). All results are shown in Table 9. It is obvious that SAPA consistently improves the performance under all settings.

## D.3    HYPERPARAMETER

The radius $\rho$ in Eq. 8 is the most important hyperparameter in our proposed method, and it represents the radius within which the locally worst model is searched. In the work (Foret et al., 2020), the default value is 0.05 and we adopt it in our main experiments. However, we are interested in

Table 9: Avg. Success Rate under the "Multiple-Victim" Setting in Targeted Attacks.

| | | 16/255 | | 8/255 | | 4/255 | |
|---|---|---|---|---|---|---|---|
| | | 1% | 0.25% | 1% | 0.25% | 1% | 0.25% |
| CIFAR10: 4 Victims | Grad-Match | 62.9 | 36.2 | 34.7 | 25.1 | 20.3 | 7.5 |
| | SAPA | **75.1** | **53.4** | **47.9** | **30.8** | **24.3** | **10.8** |
| CIFAR10: 8 Victims | Grad-Match | 52.1 | 23.2 | 27.9 | 18.4 | 12.7 | 5.6 |
| | SAPA | **64.6** | **31.2** | **34.6** | **26.1** | **17.5** | **7.3** |
| CIFAR100: 4 Victims | Grad-Match | 67.3 | 30.1 | 37.2 | 12.5 | 17.8 | 2.9 |
| | SAPA | **74.2** | **43.8** | **44.3** | **19.7** | **25.1** | **6.1** |
| CIFAR100: 8 Victims | Grad-Match | 43.6 | 23.2 | 16.7 | 4.9 | 13.8 | 2.7 |
| | SAPA | **52.7** | **31.3** | **24.8** | **8.3** | **18.7** | **4.2** |

whether this hyperparameter has a large impact on the performance. Therefore, we test for different values, i.e. $\rho = 0.01, 0.05, 0.1, 0.2, 0.5$. To avoid the computational cost, we only test on targeted(SAPA+Grad-Match) and backdoor attack(SAPA+SleeperAgent), and results are shown in Tabel 10. We notice that a smaller $\rho$ causes the algorithm to regress to the backbone attacks without SAPA, so the poisoning effect drops; while a very large value ($\rho \geq 0.2$) also leads to poor attack performance.

Table 10: Hyperparameter $\rho$

| Method/$\rho$ | 0.01 | 0.05 | 0.1 | 0.2 | 0.5 |
|---|---|---|---|---|---|
| SAPA+GM | 76.5 | 80.4 | 82.1 | 77.5 | 69.3 |
| SAPA+SleepAgent | 92.7 | 97.1 | 97.6 | 96.8 | 83.2 |

### D.4 Ensemble and re-initialization

In Section.5.5, we compare SAPA with *ensemble* and *re-initialization* techniques. It is also worth noting that SAPA can leverage these techniques to further improve the performance. We conduct experiments on targeted(SAPA+Grad-Match) and backdoor(SAPA+SleeperAgent) attacks to show this. Same as in Section.5.5, let $K, R$ denote the number of ensembles and re-initializations respectively.

Table 11: Impact of ensemble(K) and re-inistailzation(R)

| | Targeted | | Backdoor | |
|---|---|---|---|---|
| | ASR($\uparrow$) | Time/s($\downarrow$) | ASR($\uparrow$) | Time/s($\downarrow$) |
| K=1,R=1 | 80.0 | 156.9 | 97.1 | 4031.3 |
| K=2,R=1 | 83.5 | 319.1 | 98.2 | 8129.7 |
| K=4,R=1 | 86.3 | 748.4 | 98.9 | 16254.3 |
| K=8,R=1 | 87.1 | 1562.5 | 100 | 30888.9 |
| K=1,R=2 | 81.7 | 296.8 | 97.9 | 8006.4 |
| K=1,R=4 | 83.5 | 718.5 | 98.5 | 15986.7 |
| K=1,R=8 | 86.2 | 1379.7 | 99.3 | 32459.8 |

### D.5 More results

We also provide the performance of un-targeted attacks with standard error reported, and of targeted attacks for ResNet50 on Cifar10 in Table 12 and 13 respectively.

## E Additional visualizations

**Loss Landscape**. We provide the visualization of the loss landscape for targeted and backdoor attacks to further illustrate that our method can indeed minimize the sharpness. For both attacks, losses are computed on the victim set $D_T$, and we visualize on CIFAR10 and ResNet18 with perturbation size $16/255$ and poison ration 1%. Visualizations are shown in Figure.3. It is obvious that our method can achieve a smoother loss landscape.

**Effect of epochs on targeted**. As we mentioned in Section 5, we adopt different re-training epochs when applying baseline Grad-Match. In specific, the original paper(Geiping et al., 2020) adapts

Table 12: Model Test Accuracy under Un-targeted Attacks. (with standard error reported)

| | CIFAR10 | | | | | | CIFAR100 | | | | | |
| --- | --- | --- | --- | --- | --- | --- | --- | --- | --- | --- | --- | --- |
| | ResNet18 | | | ResNet50 | | | ResNet18 | | | ResNet50 | | |
| | 100% | 80% | 50% | 100% | 80% | 50% | 100% | 80% | 50% | 100% | 80% | 50% |
| Separable | 13.5±0.11 | 86.3±0.17 | 92.9±0.26 | 14.9±0.12 | 88.1±0.17 | 93.2±0.11 | 9.14v0.16 | 57.1±0.21 | 66.2±0.19 | 8.4±0.14 | 60.8±0.17 | 66.7±0.25 |
| Autoregressive | 11.7±0.13 | 82.2±0.25 | 89.7±0.28 | 10.07±0.09 | 83.6±0.15 | 90.3±0.11 | 4.24±0.08 | 51.6±0.13 | 64.7±0.10 | 4.32±0.11 | 53.5±0.21 | 66.1±0.19 |
| Error-Max | 15.4±0.23 | 88.2±0.34 | 92.2±0.27 | 27.8±0.38 | 90.1±0.25 | 93.9±0.31 | 4.87±0.11 | 57.3±0.14 | 66.9±0.13 | 5.61±0.13 | 58.3±0.18 | 68.1+0.27 |
| Error-Max+SAPA | 11.3±0.11 | 84.6±0.23 | 90.1±0.19 | 15.76±0.34 | 85.7±0.39 | 91.3±0.23 | 4.13±0.08 | 55.1±0.12 | 64.8±0.09 | 4.87±0.17 | 56.8±0.14 | 66.9±023 |
| Error-Min | 21.2±0.26 | 87.1±0.38 | 93.4±0.35 | 18.89±0.41 | 89.5±0.36 | 94.5±0.29 | 11.2±0.19 | 56.9±0.25 | 67.7±0.17 | 10.8±0.14 | 60.5±0.21 | 70.3±0.24 |
| Error-Min+SAPA | 10.9±0.15 | 83.7±0.32 | 90.0±0.38 | 10.3±0.37 | 85.2±0.39 | 91.8±0.33 | 8.73±0.21 | 53.1±0.13 | 65.3±0.15 | 9.52±0.12 | 57.9±0.17 | 67.6±0.18 |

Table 13: Success Rate in Targeted Attacks(for ResNet50 on Cifar10).

| | | 16/255 | | 8/255 | | 4/255 | |
| --- | --- | --- | --- | --- | --- | --- | --- |
| | | $\epsilon = 1\%$ | $\epsilon = 0.2\%$ | $\epsilon = 1\%$ | $\epsilon = 0.2\%$ | $\epsilon = 1\%$ | $\epsilon = 0.2\%$ |
| CIFAR10 | Bullseye | 1.7±0.9 | 0.5±0.2 | 1.1±0.6 | 0.4±0.1 | 0±0 | 0±0 |
| | Poison-Frog | 2.3±0.7 | 0.6±0.4 | 0.8±0.3 | 0±0 | 0±0 | 0±0 |
| | Meta-Poison | 46.9±2.5 | 34.1±2.6 | 23.5±1.7 | 10.7±1.6 | 15.6±1.3 | 6.9±1.2 |
| | Grad-Match | 75.6±3.7 | 53.9±4.3 | 30.1±3.8 | 14.8±2.6 | 21.5±2.9 | 9.6±2.3 |
| | SAPA | **81.4±3.1** | **60.2±3.5** | **34.7±3.6** | **17.6±1.9** | **26.3±2.1** | **12.4±1.9** |
| CIFAR100 | Bullseye | 2.4±1.1 | 1.2±0.5 | 1.6±0.9 | 0.7±0.3 | 0±0 | 0±0 |
| | Poison-Frog | 2.9±0.9 | 1.3±0.4 | 1.5±0.7 | 0.8±0.2 | 0±0 | 0±0 |
| | Meta-Poison | 50.2±2.4 | 29.8±1.8 | 27.9±2.1 | 12.5±1.1 | 19.4±1.6 | 8.2+1.8 |
| | Grad-Match | 80.8±3.6 | 50.7±4.1 | 36.7±3.8 | 24.3±3.9 | 26.9±2.7 | 10.4±2.1 |
| | SAPA | **84.5±3.4** | **56.4±2.9** | **42.8±3.4** | **29.6±2.7** | **30.1±2.4** | **13.5±1.6** |

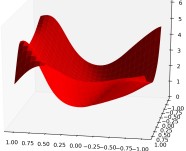 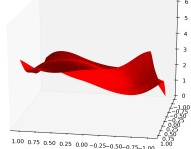 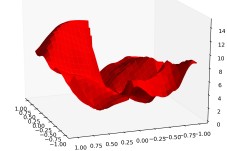 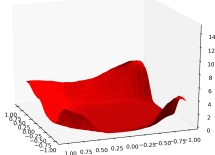

(a) Backdoor(SleeperAgent)  (b) Backdoor(SAPA)  (c) Targeted(Grad-Match)  (d) Targeted(SAPA)

Figure 3: Visualization of loss landscape for Backdoor and Targeted attacks.

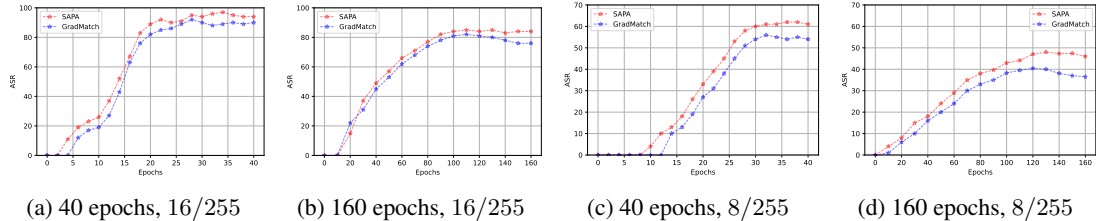

(a) 40 epochs, 16/255  (b) 160 epochs, 16/255  (c) 40 epochs, 8/255  (d) 160 epochs, 8/255

Figure 4: Attack successful rate (ASR) for different training epochs of targeted attacks

40 epochs while we re-trained for 150 epochs. We plot the success rates for different epochs(40 and 160) under different perturbation sizes(16/255 and 8/255) and show them in Figure 4. Results of SAPA and Grad-Match are in red and blue respectively. From the figures, we notice that the 160-epoch has a smaller success rate than the 40-epoch and the success rate is slightly decreasing when the epochs are growing. These observations imply that re-training epochs indeed influence the poisoning effect and SAPA can improve the performance to some extent.

