# OpenReview forum: "Sharpness-Aware Data Poisoning Attack"
_ICLR.cc/2024/Conference — ICLR 2024 spotlight_

### Official Review · Reviewer_Twq5 · 2023-10-24

**Soundness:** 3 good
**Presentation:** 3 good
**Contribution:** 2 fair
**Rating:** 6
**Confidence:** 4

**Summary:**

In this paper, the authors propose a new approach, called Sharpness-aware Data Poisoning Attack (SAPA), that use loss landscape sharpness to improve the effect of data poisoning attack under the worst scenario (model re-training). The authors elaborate on achieving SAPA in targeted, untargeted and backdoor attacks. SAPA can be incorporated with existing data poisoning approaches and achieve better performance. Evaluations are performed on CIFAR-10 and CIFAR-100 over different tasks (targeted,  untargeted, backdoor) and the results show that SAPA yields better ASR compared to existing attacks.

**Strengths:**

1. This paper proposes an effective approach to tackle the long-last issue that poisoning attack effect may drop significantly due to worst-case training settings (e.g., re-training).

2. The proposed SAPA is applicable to multiple attacks (i.e., targeted, untargeted, backdoor).

3. The authors perform comprehensive experiments and ablations to show the performance of SAPA and the results are sufficient to support their claim that SAPA can enhance ASR by a large margin.

**Weaknesses:**

1. SAPA aims to solve the effectiveness of certain attacks (e.g., clean-label integrity and backdoor attacks) that suffer from re-training. On the other side, there existing a line of attacks (e.g., dirty-label attacks such as LIRA, WaNet, BadNets, etc.) that do not have such performance drop. I would suggest the authors clarify the discrepancy between different attacks.

2. The authors only use CIFAR-10 and CIFAR-100 in the evaluations. However, in prior works such as the Hidden trigger backdoor and Gradient matching, ImageNet is always used to evaluate the attack performance. I suggest the authors also include ImageNet results in the experiments.

3. It would be desired to evaluate SAPA against SOTA defenses. The authors provide the result against adversarial training, which is not the best option to defend against poisoning attacks. There are many defenses specifically designed for poisoning attacks such as ABL [1], Fine-pruning [2], Deep KNN[3], etc, it would be nice to evaluate SAPA against these defenses.

[1]. Anti-Backdoor Learning: Training Clean Models on Poisoned Data

[2]. Fine-Pruning: Defending Against Backdooring Attacks on Deep Neural Networks

[3]. Deep k-NN Defense against Clean-label Data Poisoning Attacks

**Questions:**

Please see weaknesses for details.

---

> ### Author Response · Authors · 2023-11-17
> **Response to Reviewer Twq5**
>
> We thank the reviewer for raising these concerns, and we will address them by discussing the following questions: (1) clarify the discrepancy between different attacks; (2) evaluations on the ImageNet dataset; and (3) Evaluate SAPA against SOTA defenses.
>
> **Q1: Clarify the discrepancy between different attacks**
>
> We thank the reviewer for pointing out another line of attacks that do not suffer from the re-training. We add descriptions of these attacks and clarify the discrepancy between them and our studied attacks in Appendix B.2, and refer to them in Section 2.1.
>
> We also provide some introductions here. Some of these attacks mentioned by the reviewer, such as WaNet and LiRA, need additional assumptions that the attacker can control the re-training process. This is different from our considered "end-to-end" scenario as described in Section 2.1. Besides, BadNet attack does not generate poisoning samples targeting on specific models. Thus, it would not encounter the issue of re-training uncertainty. However, this attack typically considers inserting noticeable triggers.
>
> **Q2: Evaluations on the ImageNet dataset.**
>
> We thank the reviewer for this valuable suggestion. We conduct experiments on targeted and backdoor attacks. In detail, we test on ResNet18 and budget $\epsilon=4/255,8/255,16/255$, poison rate $p=0.1\\%, 0.05\\%$ for targeted attacks; test on ResNet18, ResNet50, VGG11, budget $\epsilon=8/255, 16/255$ and poison rate $p=0.05\\%$ for backdoor attacks. We present the success rate of Grad-Match and Grad-Match+SAPA for targeted attacks; and the success rate of Sleeper-agent and Sleeper-agent+SAPA for backdoor attacks for convenient comparison in the following Tables. We will conduct experiments on other attacks and put them in the appendix. According to the results, our method consistently improves the performance of the backbone method on the large dataset.
>
> | targeted ASR(↑)   | ε = 16 |        | ε = 8  |        | ε = 4  |        |
> |----------------|--------|--------|--------|--------|--------|--------|
> | Attacks         | p = 0.1% | p = 0.05% | p = 0.1% | p = 0.05% | p = 0.1% | p = 0.05% |
> | Grad-Match     | 91.5   | 77.6   | 82.4   | 68.2   | 58.3   | 45.3   |
> | Grad-Match + SAPA | 93.4   | 80.6   | 86.2   | 74.1   | 69.1   | 55.7   |
>
> | backdoor ASR(↑)   | ResNet18 |     | ResNet50 |    | VGG11  |     |
> |-----------------|-------------|---|-------------|---|-----------|---|
> | Attacks          |ε = 16        |ε = 8  | ε = 16  | ε = 8  | ε = 16  | ε = 8  |
> | Sleeper-agent  | 46.2   | 37.4   | 39.7   | 27.6   | 41.8   | 34.5   |
> | Sleeper-agent + SAPA | 54.7   | 42.8   | 48.6   | 35.1   | 53.8   | 41.2   |
>
> **Q3: Evaluate SAPA against SOTA defenses.**
>
> We thank the reviewer for pointing out these defenses. We conduct experiments on targeted attacks against Deep-KNN[1] and backdoor attacks against ABL [2], Fine-pruning [3]. For both attacks, we test on CIFAR-10 with $\epsilon=16/255$ and $p=1\\%$. Results(success rate) for Deep-KNN are shown in the first Table below and results(success rate) for ABL and Fine-pruning are in the second Table below. According to the results, Deep-KNN can hardly defend our attacks and our method can consistently boost the performance of backbone methods against powerful defenses such as ABL and FP.
>
> |     ASR(↑)       | No defenses | Deep-KNN |
> |----------------|-------------|----------|
> | Bullseye       | 3.7         | 0.9      |
> | Poison Frogs   | 1.3         | 0.7      |
> | Grad-Match     | 72.9        | 70.5     |
> | Grad-Match+SAPA| 80.1        | 79.4     |
>
> |    ASR(↑)       | no defenses | ABL  | FP   |
> |----------------|-------------|------|------|
> | Hidden-trigger | 3.5         | 1.7  | 2.4  |
> | clean-label    | 2.7         | 0.8  | 2.1  |
> | sleeper-agent  | 90.9        | 57.4 | 59.1 |
> | sleeper-agent+SAPA | 98.1    | 63.7 | 65.8 |
>
> We hope our responses can address your concerns. We appreciate your valuable reviews and look forward to further feedback.
>
> **References**
>
> [1] Deep k-NN Defense against Clean-label Data Poisoning Attacks
>
> [2] Anti-Backdoor Learning: Training Clean Models on Poisoned Data
>
> [3] Fine-Pruning: Defending Against Backdooring Attacks on Deep Neural Networks

---

> > ### Comment · Reviewer_Twq5 · 2023-11-18
> > **Thank you for your response.**
> >
> > The authors addressed my concerns adequately. I'd like to keep my rating.

---

> > > ### Author Response · Authors · 2023-11-20
> > >
> > > We are glad to hear that! Thank you for your feedback.

---

### Official Review · Reviewer_kdwN · 2023-10-29

**Soundness:** 3 good
**Presentation:** 3 good
**Contribution:** 3 good
**Rating:** 6
**Confidence:** 5

**Summary:**

This paper proposes a data poisoning attack by leveraging sharpness-aware minimization. The method can be applied to backdoor, untargeted, and targeted attacks to improve performance.

**Strengths:**

1. This paper introduces a unique approach using loss landscape sharpness to enhance poisoning attacks.

2. The method can be applied to various attack settings.

3. The performance shows the proposed method is effective against existing defenses.

**Weaknesses:**

The authors have improved the paper from their NeurIPS submission. The remaining concern is that many of the reported results are still not consistent with the results presented in the original papers without proper justification, including both Table 1 and Table 2.

**Questions:**

See weaknesses.

---

> ### Author Response · Authors · 2023-11-17
> **Response to Reviewer kdwN**
>
> We thank the reviewer for raising this concern. We would like to clarify why some baselines are lower than reported in the original paper. First, as mentioned in the footnote of Table 1, the original paper of Grad-Match only trains 40 epochs while we train for 160 epochs, and we include a detailed analysis of the impact of epochs in Appendix D. Second, as we mention in the footnote of Table 2, Hidden trigger and Clean-label are originally proposed for fine-tuning settings (by fixing the early layers), while we focus on an end-to-end setting in this work, and our results are consistent with results in work [1].
>
> We hope our responses can address your concerns. We appreciate your valuable reviews and look forward to further feedback.
>
> [1] Sleeper agent: Scalable hidden trigger backdoors for neural networks trained from scratch.

---

> ### Author Response · Authors · 2023-11-20
> **A friendly reminder**
>
> We are grateful for your reviews. We hope that our responses have addressed your concerns. If you have any further concerns, please let us know. We are looking forward to hearing from you.

---

> > ### Comment · Reviewer_kdwN · 2023-11-23
> > **Thanks for the authors' response!**
> >
> > I appreciate the authors' response, and I am inclined to maintain my score.

---

> > > ### Author Response · Authors · 2023-11-23
> > >
> > > Thank you for your feedback and support!

---

### Official Review · Reviewer_X4g5 · 2023-10-30

**Soundness:** 3 good
**Presentation:** 4 excellent
**Contribution:** 3 good
**Rating:** 6
**Confidence:** 4

**Summary:**

This work proposes a modification to the objective function of several poisoning attacks wherein poisons are crafted at a parameter vector that has been modified from a standard (ERM trained parameter vector) to increase the attacker's target loss.

**Strengths:**

1. The authors do a good job of comparing to a wide range of existing attacks in several poisoning settings.

2. The authors also do thorough experimentation in each of these settings, and include some results under existing defenses.

3. Taking the results of the work at face value, the proposed method improves over SOTA poisoning methods, sometimes by a significant margin.

4. The paper is easy to follow and generally well presented.

**Weaknesses:**

1. I think the motivation of poisoning the "worst" retrained model isn't fleshed out enough. Why does crafting on the worst-case poisoned model intuitively lead to more generalizable/potent poisons on *average*?

  2. The authors should be more careful when talking about sharpness of minima in this context, as sharpness depends on the *objective* in question. Many might read this work and assume the sharpness the authors are referring to is the sharpness of the minima of the "standard" loss landscape, when this is not what is being discussed.

3. I would avoid statements like "with a high possibility, the re-trained model can converge to a point where the poisoning effect presents" , and "Therefore, for the models which are re-trained on the poisoned dataset, the poisoning effect is also very likely to persist" as these haven't been justified anywhere in the work.

4. A concern raised about existing attacks  in the introduction was that they are not architecture agnostic. While I agree with this concern, this work does nothing to explicitly mitigate this.

5. Figure 2 shows that when using ensembles, and restarts, existing methods can beat SAPA. Did you try SAPA with these additions? In general I'm not sure poison crafting time is something to include in the main body/claim as an advantage, especially when the worst times are ~20 minutes.

6. Gradient matching also considers ensembles.

**Questions:**

1. Does SAPA also improve poisoning success when paired with objectives other than gradient matching?

2. Eq. 7 is unclear - you craft $D_p$ on at a parameter vector $\theta^* + \nu$, but $\theta^*$ was the result of ERM on $D_{tr} + D_p$? How was the initial $D_p$ (used in ERM) calculated? Do you have to generate poisons/train a victim model twice?

---

> ### Author Response · Authors · 2023-11-17
> **Response to Reviewer X4g5 (1/2)**
>
> We thank the reviewer for raising these concerns. We will address them by discussing the following questions: (1) Why does crafting on the worst-case poisoned model intuitively lead to more generalizable/potent poisons on average; (2) the reference of the term "sharpness" in this work; (3) discussion about some statements mentioned by the reviewer; (4) does our work explicitly mitigate the concern that existing attacks are not architecture-agnostic; (5) Gradient matching considers ensembles, can SAPA be combined with ensembling and restarts (6) can SAPA improve other objectives other than gradient matching; and (7) clarification for Eq.7.
>
> **Q1: Why does crafting on the worst-case poisoned model intuitively lead to more generalizable/potent poisons on average?**
>
> We would like to provide some intuition about the "worst" re-trained model below. As we mentioned in Section 4.1, the **worst-case poisoned model** is the model that has the worst poisoning effect among the possible models that are trained on the poisoned dataset (considering various training uncertainties including initialization, training algorithm, etc.). In other words, the poisoning effect on the worst-case model serves as the lower bound of poisoning effects for all models. Therefore, in the re-training stage, the poisoning effect of the real victim model should also be better than this worst-case model, so by maximizing the poisoning effect on the worst-case model we can guarantee that a high poisoning effect is preserved for other models.
>
> **Q2: The reference to the term "sharpness" in this work.**
>
> We thank the reviewer for pointing out the difference between the sharpness in this work and the sharpness of the "standard" loss landscape. We refine our presentation to avoid potential misunderstanding about the concept of sharpness in the revised paper. Specifically, we clarify this in the paragraph above Eq.(8) where the sharpness is defined.
>
> **Q3: Discussion about some statements mentioned by the reviewer**
>
> We agree that we should be more careful of these statements. We remove the unjustified statements such as the first one. For the second statement, our experiments can provide some empirical justifications. According to our experimental results in Table 1-3, SAPA improves the attacking performance of backbone methods, indicating that more poisoning effects are preserved. The curve of the poisoning effect in Figure 1 also provides empirical evidence that SAPA leads to better poisoning effects.
>
> **Q4: Does our work explicitly mitigate the concern that existing attacks are not architecture-agnostic?**
>
> We agree that we do not devise explicit strategies to address the uncertainty due to model architecture. Thus, we correspondingly emphasize this fact at the end of Section 4.1 in our paper.
>
> Besides, according to our experimental results in Table 4 where poisons are generated from one model and tested on models with different architectures, our method can improve the poisoning effect of backbone methods. This result suggests that SAPA could also implicitly bring improvements under the model architecture variation.
>
>
> **Q5: Gradient matching considers ensembles, can SAPA be combined with ensembling and restarts?**
>
> We are aware that Gradient matching considers ensembles in the original paper. Thus, in our paper, we have also compared our method with Grad-match with ensembles in Section 5.5 and Figure 2. In detail, we compare SAPA with Grad-match incorporated with different setups of Ensembling and Re-initialization. From the result, we can find that our method can achieve comparable performance to the Grad-matching with a large number of ensembled models, while our method has a much better time efficiency.
> As a conclusion, we believe that time efficiency is an advantage of our method. As shown in Figure 2, the ensembled Grad-match needs much more time than our method (without ensembles) to achieve a similar performance.
>
> We also acknowledge that SAPA can be combined with Ensembling and Re-initialization (E&R). Thus, in Appendix Table 11, we have conducted experiments to apply E&R to SAPA on both targeted and backdoor attacks. According to these results, E&R can further improve the performance of SAPA while with the cost of time efficiency.
>
> **Q6: Can SAPA improve objectives other than gradient matching?**
>
> Yes, apart from Grad-match, we also combine SAPA with objectives of un-targeted attacks, i.e. Error-min in Section 4.3 and Error-max in Appendix C.3. We conduct experiments for both objectives and show them in Table 3. We observe obvious improvement for both attacks, and this further illustrates the wide range of applicability of SAPA.

---

> ### Author Response · Authors · 2023-11-17
> **Response to Reviewer X4g5 (2/2)**
>
> **Q7: Clarification for Eq.7.**
>
> Eq.7 serves to approximate the worst-case adversarial loss utilizing the concept of sharpness with respect to a given poisoning set $D_p$and an adversary set $D_T$. We would like to point out that our poison crafting does not directly utilize Eq.7; instead, the actual optimization problem we engage with is formulated in Eq.8. This is where the bi-level optimization problem is defined, leading us to the generation of $D_p$.
>
> As detailed in Algorithms 1-4, the initial poisoning set $D_p$ starts with a random initialization and is iteratively updated during the generation process. Thus we do not need to generate them twice. The updating of $\theta^*$ depends on the backbone methods SAPA is combined with. For SAPA+Grad-match, $\theta^*$ is the model trained on clean data and fixed during the generation which follows the original algorithm in [1]; for SAPA+Error-min and SAPA+Error-max, $\theta^*$ is also randomly initialized and iteratively updated as shown in Algorithm 1 and 4.
>
> [1] Witches' Brew: Industrial Scale Data Poisoning via Gradient Matching

---

> ### Author Response · Authors · 2023-11-20
> **A friendly reminder**
>
> We are grateful for your reviews. We hope that our responses have addressed your concerns. If you have any further concerns, please let us know. We are looking forward to hearing from you.

---

### Official Review · Reviewer_jHfE · 2023-11-01

**Soundness:** 3 good
**Presentation:** 3 good
**Contribution:** 3 good
**Rating:** 6
**Confidence:** 3

**Summary:**

Previous studies have developed different types of data poisoning attacks, but they are limited by the uncertainty of the retraining process after the attack. To address this challenge, the paper proposes a novel method called Sharpness-Aware Data Poisoning Attack (SAPA), which leverages the concept of deep neural networks' (DNNs') loss landscape sharpness to optimize the poisoning effect on the worst possible retrained model. Extensive experiments are conducted to show the method's effectiveness.

**Strengths:**

1. The idea of leveraging the concept of loss landscape sharpness to improve data poisoning's efficiency is intriguing. The proposed method is also applicable to different attack objectives.
2. The paper includes vastly extensive experiments with different attack goals, model architectures, re-training variants, etc.
3. Overall, the experimental results show this method could constantly outperform other baselines.
4. The paper is quite well-writtenm. Especially, the proposed method is explained quite clearly and detailedly.

**Weaknesses:**

1. The attack success rates are quite unimpressive when the perturbation budgets are low (4/255 for targeted attack and 8/255 for backdoor attack). I recognize there are improvements compared to other baselines, but I find it quite hard to considered the attacks are successful.
2. The number of defense strategies evaluated in the paper is quite lacking. All defense strategies considered are different re-training variants. While the adversary in this theat model acts as the data provider, I think there should be experiments with data filtering defenses, such as [1], [2], [3], [4]. There is also no poisoned model detection defense, such as Neural Cleanse or STRIP, mentioned in the paper.
3. The performance against adversarial training is also quite underwhelming.

[1] Chen, Bryant, et al. "Detecting Backdoor Attacks on Deep Neural Networks by Activation Clustering." (AAAI 2019)
[2] Tran, Brandon, Jerry Li, and Aleksander Madry. "Spectral signatures in backdoor attacks." (NeurIPS 2018)
[3] Hayase, Jonathan, et al. "Spectre: Defending against backdoor attacks using robust statistics." (ICML 2021)
[4] Zeng, Yi, et al. "Rethinking the backdoor attacks' triggers: A frequency perspective." (ICCV 2021)

**Questions:**

1.  Regarding my concern about evalutation against more defense approaches, I would recommend adding experiments with some of the aforementioned defenses
2. It would be better if there are qualitative comparison between clean and poisoned images.
3. I find this sentence in section 5.5 confusing: "In this part, we compare the efficiency and effectiveness trade-off between SAPA and E&R, when they are incorporated to existing defenses, such as Gradient Matching and Sleeper Agent." Do the authors mean "existing attacks" instead?
4. Table 3 is quite hard to comprehend since it does not have best performances highlighted or average accuracy drops. The authors could improve its presentation a bit.

---

> ### Author Response · Authors · 2023-11-17
> **Response to Reviewer jHfE**
>
> We thank the reviewer for raising these concerns. We will address these concerns by discussing the following questions: (1) Why the success rate is low when the budget is low and how to improve it; (2) evaluation on more defenses; (3) why the performance against adversarial training is not good enough; and (4) additional comments.
>
> **Q1: Why the success rate is low when the budget is low?**
>
> We thank the reviewer for raising this concern and we agree that it is more difficult to conduct attacks when the budget is very low. However, a low perturbation budget could possibly make the attacks become more unnoticeable in practice. Thus, there is a chance that the attacker can correspondingly increase the poisoning rate for a better poisoning effect. As illustrated in the following Table, by increasing the poisoning rate to 5%, we observe a significant enhancement in the success rate, exceeding 30%.
>
> | Targeted, $\epsilon$=4 | Ciaffr10 |      |      |
> |---------------------|----------|------|------|
> |                     | p=5%     | p=1% | p=0.2% |
> | Grad-Match          | 23.1     | 10.3 | 6.4  |
> | SAPA+Grad-Match     | 34.7     | 16.7 | 11.2 |
>
> Therefore, these unnoticeable attacks, though have low average success rates, can still be harmful. Moreover, existing works in poisoning attacks [10][11][12] also consider this scenario, and our work follows the same setting. The experimental results have shown that our method outperforms them, as seen in Table 1-2 in our paper.
>
> **Q2: Evaluation on more defenses**
>
> We thank the reviewer for pointing out these defenses, and we conduct experiments to evaluate our method. We test on the CIFAR-10 dataset with ResNet18. We notice that these defenses are designed for backdoor attacks, so we conduct on backdoor attacks only and fix the budget $\epsilon=16/255$ and poison rate p=1%. We report the success rate in the following Table. According to these results, our method can improve the backbone attack's resistance against various defenses.
>
> |                       | no defenses | NC[1] | SS[2] | Spectre[3] | AC[4] | Strip[5] | DCT[6] |
> |-----------------------|-------------|-------|-------|------------|-------|----------|--------|
> | Hidden-trigger        | 3.5         | 1.2   | 2.3   | 0.8        | 1.1   | 0.9      | 0.5    |
> | clean-label           | 2.7         | 1.9   | 1.5   | 0.5        | 1.7   | 0.6      | 0.3    |
> | sleeper-agent         | 90.9        | 83.9  | 35.5  | 29.6       | 15.2  | 62.7     | 42.8   |
> | sleeper-agent+SAPA    | 98.1        | 87.2  | 44.8  | 38.1       | 29.3  | 70.1     | 49.3   |
>
> **Q3: Why the performance against adversarial training is not good enough?**
>
> We admit that the performance against adversarial training is not good enough in Table 5. In fact, adversarial training is a powerful defense against un-targeted attacks, and previous works such as [7][8][9], also suffer from this defense. According to our experiments in Table 5, our method can improve the resistance of backbone attacks and outperform all the baselines. It remains a challenge to break the defense of adversarial training and we will keep exploring in this direction.
>
> **Additional comments**
>
> We really appreciate the reviewer's suggestions. We will include evaluations on defenses in Q2, as well as image visualizations, in the appendix. We apologize for the typos in that sentence in Section 5.5, which should be 'existing attacks'. We have carefully revised the paper and ensured no other typos. We also revise Table 3 to make it more readable.
>
> **References**
>
> [1] Neural Cleanse: Identifying and Mitigating Backdoor Attacks in Neural Networks
>
> [2] Spectral signatures in backdoor attacks
>
> [3] Spectre: Defending against backdoor attacks using robust statistics.
>
> [4] Detecting Backdoor Attacks on Deep Neural Networks by Activation Clustering.
>
> [5] STRIP: A Defence Against Trojan Attacks on Deep Neural Networks
>
> [6] Rethinking the backdoor attacks' triggers: A frequency perspective.
>
> [7] Unlearnable Examples: Making Personal Data Unexploitable
>
> [8] Adversarial Examples Make Strong Poisons
>
> [9] Autoregressive Perturbations for Data Poisoning
>
> [10] Witches' Brew: Industrial Scale Data Poisoning via Gradient Matching
>
> [11] MetaPoison: Practical General-purpose Clean-label Data Poisoning
>
> [12] Poison frogs! targeted clean-label poisoning attacks on neural networks

---

> ### Author Response · Authors · 2023-11-20
> **A friendly reminder**
>
> We are grateful for your reviews. We hope that our responses have addressed your concerns. If you have any further concerns, please let us know. We are looking forward to hearing from you.

---

### Official Review · Reviewer_Ae7N · 2023-11-01

**Soundness:** 4 excellent
**Presentation:** 3 good
**Contribution:** 3 good
**Rating:** 6
**Confidence:** 4

**Summary:**

The paper induces sharpness-aware training method towards 3 poisoning tasks: targeted attack (perturb few data to misclassify a sample), backdoor attack (perturb few data to misclassify a class of samples), and unlearnable examples (perturb all data to misclassify all clean test samples). The key design is an additional step to calculate the worst poisoning model before using it to update the poison samples. Experiments show that when plugged in to existing methods, SAPA improves the poison performance steadily.

**Strengths:**

1. The paper extensively studies three poison attacks to showcase the effectiveness of SAPA. It would be very helpful for the audience since most papers only focus on one task but term the task as poisoning.
2. The introduction of the sharpness-aware idea to poison attacks is straightforward and easy to plug in by adding an adversarial loop. But please make the additional computation amount (X + SAPA v.s. X) more clear.
3. Experiments are comprehensive to demonstrate the steady improvement by SAPA, even under various training strategies. The study on efficiency is informative.

**Weaknesses:**

1. It would be good if the authors clearly distinguish between different poison tasks before introducing the method. Currently, the threat model is not clear for 3 tasks, and it may be confusing to distinguish the contribution of SAPA in a specific task.
2. To better demonstrate the plug-in property of SAPA, it is good to be described without a specific attack method, e.g., gradient matching or error-min. And the results should be shown as X v.s. X+SAPA also in targeted and backdoor attacks.
3. "Self-Ensemble Protection: Training Checkpoints Are Good Data Protectors" also focuses on efficient and effective poisoning in retraining. How does SAPA compare to it in the unlearnable examples task in terms of attack performance and efficiency?

**Questions:**

Response to rebuttal: Thanks for the good rebuttal and revision of the paper. I have no future concerns and thus keep my score.

---

> ### Author Response · Authors · 2023-11-17
> **Response to Reviewer Ae7N**
>
> We thank the reviewer for raising these valuable comments. In rebuttal, we provide clarifications on the following issues: (1) the threat models for different attacks; (2) a general description for SAPA; and (3) compare our work with another existing work.
>
> **Q1: Clarification for the threat models.**
>
> We follow the reviewer's suggestion and add details about the threat model for each type of attack, including the attacker's goal, attacker's capacity, and adversary objectives. We put these details in Appendix B.1 and refer to them in Section 2.1.
>
> **Q2: A general description of SAPA.**
>
> We thank the reviewer for giving this suggestion. We follow the reviewer's suggestions and modify the names of methods in the experiment part.
>
> In Eq.(8) in section 4.1, we define a general objective for SAPA, where we replace the original adversarial loss with a sharpness variant to approximate the local worst-case loss. Due to the distinctions between backbone algorithms, we also separately discuss the implementation of SAPA for different attacks, shown in Section 4.2, 4.3 and Appendix C.2 respectively.
>
> **Q3: Compare our work with another existing work.**
>
> We thank the reviewer for pointing out the recent method [1] in un-targeted attacks. According to Table 4 and Section 4.4 in the original paper [1], SEP can achieve better performance than the ensembled error-max (referred to as AdvPoison in [1]) while having better time efficiency. In our rebuttal, we also present a direct comparison of SEP, Error-max and SAPA+Error-max, Error-min and SAPA+Error-min under $\epsilon=8/255$ in the following Table. The results of SEP and SAPA are directly collected from the paper [1] and our paper respectively.
>
> | Test Acc (↓)     | CIFAR-10 | CIFAR-100 |
> |-------------------|----------|-----------|
> | SEP               | 4.7      | 2.6       |
> | Error-max         | 11.9     | 4.8       |
> | SAPA+Error-max    | 9.6      | 4.1       |
> | Error-min         | 21.2     | 11.2      |
> | SAPA+Error-min    | 10.9     | 8.7       |
>
> According to these results, SEP can have the best performance in the studied methods. However, we would like to emphasize that SAPA is designed as a general strategy that can enhance existing attacks. As shown in this Table, SAPA improves the performance of backbone methods and reduces the gap between them and SEP.
>
> Moreover, we notice that it is also possible to combine SAPA and SEP for further improvement. For instance, we can replace the original loss for each checkpoint in Equation 3 in [1] with a sharpness-aware loss. We are studying this problem and we will share the result if time permits.
>
> [1] Self-Ensemble Protection: Training Checkpoints Are Good Data Protectors

---

> ### Author Response · Authors · 2023-11-20
> **A friendly reminder**
>
> We are grateful for your reviews. We hope that our responses have addressed your concerns. If you have any further concerns, please let us know. We are looking forward to hearing from you.

---

### Official Review · Reviewer_Y6BU · 2023-11-02

**Soundness:** 3 good
**Presentation:** 2 fair
**Contribution:** 2 fair
**Rating:** 5
**Confidence:** 5

**Summary:**

This paper propses revisit exisitng "end-to-end" data poisoning attacks, finds that existing data poisoning attacks suffers from uncertanty issues in poison effects during the re-training process.  Then the reviewer proposes to leverage existing study on loss lanscape for DNNs and propose sharpness-award data poisoning attacks. Specifically, the authors improve upon previous work (e.g., Grad-Match, etc) with replacing original gradients with sharpness-aware gradients. Such sharpness aware loss is calculated by previous work. Through extensive experiements, sharpness-aware poisoning attackd can lead a mild improvement compared with existing approach.

**Strengths:**

1. The method is intuitive and sound.

2. The evaluation is comprehensive.

3. The results are good.

**Weaknesses:**

1. The presentation needs improvement.

2. Limited Novelty. The only contribution for this work is that the author combines sharpness-award loss function (proposed by previous work) with existing poisoning approach (e.g., Grad-Match) to make stablize the pisoning effects during the re-training peocess.

3. Lack of theoretical analysis compared with previosu work on data poisoning attacks, such as witches brew (ICLR 2020)

**Questions:**

Can you evaluate the transferability of poison effects, such as crafting poisoning samples in VGGNet but used for training ResNet.  Can you test your approach across different surrogated models and target models?

---

> ### Author Response · Authors · 2023-11-17
> **Responses to Reviewer Y6BU**
>
> We thank the reviewer for raising these concerns. We have carefully revised our paper and refined the presentation.
> In the rebuttal, we will provide clarifications on (1) the novelty of our work; (2) the relation between our work and the theoretical analysis in [2]; and (3) do we have experiments on the transferability of poison effects.
>
> **Q1: The novelty of our work.**
>
> We agree with the reviewer that our proposed method applies an existing algorithm in [1] to tackle the loss-landscape
> sharpness. However, we provide a new perspective on leveraging sharpness to solve one of the critical challenges in data poisoning attacks. In detail:
> * Sharpness [1] was previously used to improve the generalization ability of models.
> * We focus on the uncertainty problem of poisoning attacks during the re-training process. Different from existing poisoning attacks, we aim to generate poisoning data with maximum poisoning effect for the approximately "worst-case" model. In our paper, we find that the problem can be solved by the existing sharpness algorithm to find the approximate worst-case model under uncertainty. We believe this is a new perspective of applying sharpness to solve a new problem.
>
> Besides, our proposed method SAPA also differs from the existing sharpness method in terms of formulation. Existing sharpness-aware methods maximize the training loss via weight perturbation to improve the generalization of models. Differently, SAPA maximizes the loss of poisoning objective (see Eq.6 and Eq.7) to find the (locally) worst-case models via weight perturbation. This differs from the existing notion of sharpness discussed in previous studies like [1].
>
> In summary, this paper adapts an existing concept of sharpness from a new perspective for solving a totally new problem. Based on our experiments, we validate the effectiveness and efficiency of our strategy in solving our problem for various types of poisoning attacks.
>
> **Q2: The relation between our work and the theoretical analysis in [2].**
>
> We thank the reviewer for mentioning the theoretical analysis in [2]. Proposition 1 in [2] states that under some conditions such as Lipschitz continuous gradient and the alignment between gradients of training loss and adversarial loss, the victim model will converge to a stationary point of the adversarial loss. We believe that a similar analysis could be conducted on SAPA when combining SAPA and Grad-match. Compared to the original Grad-match, SAPA only replaces the model $\theta^*$ in the original adversarial loss function $Q$ with a perturbed version, $\theta^*+v$, to derive the sharpness variant $Q^S$, as detailed in Eq.10-12, while maintaining other aspects unchanged. This will not violate the original assumption in [2], such as Lipschitz continuous gradients and loss alignment. Therefore, under similar assumptions as [2], it is also potential to show that the victim model converges to a stationary point of our adversarial loss.
>
> Besides, we would like to respectfully note that a detailed convergence analysis of SAPA within the context of gradient matching falls outside the primary scope of our current work. Our focus in this paper is on establishing a general strategy applicable across various attacking scenarios, including targeted, un-targetd and backdoor attacks. Thus, it would be difficult to theoretically study the analytical behavior of our method in each of these attacks.
>
> **Q3: Do we have experiments on the transferability of poison effects?**
>
> We would like to politely point out that we present the analysis of transferability in the main text. As shown in Table 4, we employed ResNet18 as the surrogate model to generate poisons, subsequently testing them across a variety of victim models, including MobileNetV2, VGG11, and ViT. The results demonstrate that SAPA consistently enhances the performance of underlying methods across diverse attack scenarios. This performance improvement indicates the ability of SAPA to improve the transferability of attacks.
>
> We hope our responses address your concerns. We appreciate your reviews and look forward to further feedback.
>
> **References**
>
> [1] Sharpness-Aware Minimization for Efficiently Improving Generalization
>
> [2] Witches' Brew: Industrial Scale Data Poisoning via Gradient Matching

---

> > ### Comment · Reviewer_Y6BU · 2023-11-23
> > **Thanks for your response.**
> >
> > Dear Authors,
> >
> > Thanks for your response.  It addresses some of my concerns. However, i still think the novelty of this work is limited and below the bar of ICLR. Thus I raise my score to 5.

---

> > > ### Author Response · Authors · 2023-11-23
> > > **Response to Reviewer Y6BU**
> > >
> > > We sincerely thank the reviewer for raising the score and truly respect your opinion. In this response, we are delighted to provide some additional information, especially about the comparison of our work with existing literature. It may better highlight the unique contribution of our work compared to existing studies.
> > >
> > > **1. Our method vs. Existing works.** To handle the uncertainty problem of poisoning attacks during re-training, there are existing efforts before our paper (please refer to the "introduction" and "related work" sections in our paper).
> > > In specifics, the most widely-used strategy is called the "ensemble and re-initialization" (E&R) method [1,2,3], which searches for poisoning samples by considering the average poisoning effect of multiple initializations of poisoning samples and victim models. Therefore, the E\&R method faces tremendous efficiency issues. It is almost infeasible to be applied in un-targeted attacks, due to the large number of poisoning samples to be searched. **Compared to this method, we significantly improve the effectiveness-efficiency trade-off**. For example, as shown in Figure 2 in the main paper, in the case of targeted attacks, to reach a performance level comparable to that of our method, the E&R strategy requires more than **four times** of the running time. Similarly, for backdoor attacks, the E\&R approach needs **three times** of the running time to achieve similar performance outcomes as our method. In un-targeted attacks, E\&R needs more than 1 day to craft poisons on the CIFAR10 dataset.
> > >
> > > **2. New formulation.**  Our work is not a simple combination of "sharpness" and poisoning attack. In detail, our work considers the "worst-case" model (as shown in Eq.5-6) which serves as the lower bound of the poisoning effect for the models under uncertainty. Therefore, we propose to maximize the poisoning effect on the 'worst-case' model to ensure that a high poisoning effect is preserved for other models. To search this "worst-case" model, **we devise an algorithm with a similar spirit of sharpness, but a very different formulation** (please see Section 4.2 and the response to Q2 in our rebuttal).
> > > We believe this key idea of our work is novel as no previous work ever considered the uncertainty problem from this perspective.
> > >
> > > **3. Strong and broad performance improvement.** Our experiments also illustrate that our method can significantly improve the performance of different poisoning attack methods. For example, in the results of targeted, un-targeted and backdoor attacks in Table 1-3 in our paper, we show our method consistently improves the backbone attacks. Notably,
> > > in targeted and backdoor attacks, **our method sometimes improves the attack successful rate by more than 20%.** It can also successfully help the untargeted attack like Error-Min, Error-Max to further decrease the mode accuracy.
> > > This adaptability not only underscores its capacity to augment current methods at a low cost but also suggests its potential to enhance future attack strategies. Considering these factors, we firmly believe that our method represents a novel and substantial contribution to the field.
> > >
> > > **References**
> > >
> > > [1] MetaPoison: Practical General-purpose Clean-label Data Poisoning
> > >
> > > [2] Witches' Brew: Industrial Scale Data Poisoning via Gradient Matching
> > >
> > > [3] Sleeper Agent: Scalable Hidden Trigger Backdoors for Neural Networks Trained from Scratch

---

> ### Author Response · Authors · 2023-11-20
> **A friendly reminder**
>
> We are grateful for your reviews. We hope that our responses have addressed your concerns. If you have any further concerns, please let us know. We are looking forward to hearing from you.

---

> ### Author Response · Authors · 2023-11-21
> **A friendly reminder**
>
> We appreciate your reviews. We hope that our responses have adequately addressed your concerns. As the deadline for open discussion nears, we kindly remind you to share any additional feedback you may have.  We are keen to engage in further discussion.

---

### Meta-Review · Area_Chair_2LSp · 2023-12-05

**Metareview:**

This work proposes a strategy for boosting the performance of existing poisoning and backdoor attacks via sharpness-aware optimization.  They show significant performance improvements for gradient matching targeted data poisoning attacks and Sleeper Agent.  Reviewers praised the empirical results and extensive evaluations.  Some reviewers brought up the work’s limited novelty given that it combines two existing ideas (data poisoning + sharpness-aware optimization) in a straight-forward fashion, but I’m inclined to accept the paper given that no other work has combined these ideas, and their combination is highly effective.  The methodology additionally appears well communicated, so I think this paper will make a strong contribution to ICLR.

**Justification For Why Not Higher Score:**

I do not think that the methodological improvements are significant enough or that the empirical improvements are impressive enough to justify an oral.

**Justification For Why Not Lower Score:**

This work does make consistent performance improvements on an impactful performance, so I do think it warrants a spotlight.

---

### Decision · Program_Chairs · 2024-01-16

Accept (spotlight)